# Influence of Seasonality and Public-Health Interventions on the COVID-19 Pandemic in Northern Europe

**DOI:** 10.3390/jcm13020334

**Published:** 2024-01-06

**Authors:** Gerry A. Quinn, Michael Connolly, Norman E. Fenton, Steven J. Hatfill, Paul Hynds, Coilín ÓhAiseadha, Karol Sikora, Willie Soon, Ronan Connolly

**Affiliations:** 1Centre for Molecular Biosciences, Ulster University, Coleraine BT52 1SA, UK; 2Independent Researcher, D08 Dublin, Ireland; 3School of Electronic Engineering and Computer Science, Queen Mary University of London, London E1 4NS, UK; 4London Center for Policy Research, New York, NY 10004, USA; 5Spatiotemporal Environmental Epidemiology Research (STEER) Group, Environmental Sustainability & Health Institute, Technological University Dublin, D07 H6K8 Dublin, Ireland; 6Irish Centre for Research in Applied Geoscience, University College Dublin, D04 F438 Dublin, Ireland; 7Department of Public Health, Health Service Executive, Dr Steevens’ Hospital, D08 W2A8 Dublin, Ireland; 8Department of Medicine, University of Buckingham Medical School, Buckingham MK18 1EG, UK; 9Institute of Earth Physics and Space Science (ELKH EPSS), H-9400 Sopron, Hungary; 10Center for Environmental Research and Earth Sciences (CERES), Salem, MA 01970, USA

**Keywords:** COVID-19 pandemic, seasonal variation, public health, vaccination, epidemiology, Northern Europe

## Abstract

Background: Most government efforts to control the COVID-19 pandemic revolved around non-pharmaceutical interventions (NPIs) and vaccination. However, many respiratory diseases show distinctive seasonal trends. In this manuscript, we examined the contribution of these three factors to the progression of the COVID-19 pandemic. Methods: Pearson correlation coefficients and time-lagged analysis were used to examine the relationship between NPIs, vaccinations and seasonality (using the average incidence of endemic human beta-coronaviruses in Sweden over a 10-year period as a proxy) and the progression of the COVID-19 pandemic as tracked by deaths; cases; hospitalisations; intensive care unit occupancy and testing positivity rates in six Northern European countries (population 99.12 million) using a population-based, observational, ecological study method. Findings: The waves of the pandemic correlated well with the seasonality of human beta-coronaviruses (HCoV-OC43 and HCoV-HKU1). In contrast, we could not find clear or consistent evidence that the stringency of NPIs or vaccination reduced the progression of the pandemic. However, these results are correlations and not causations. Implications: We hypothesise that the apparent influence of NPIs and vaccines might instead be an effect of coronavirus seasonality. We suggest that policymakers consider these results when assessing policy options for future pandemics. Limitations: The study is limited to six temperate Northern European countries with spatial and temporal variations in metrics used to track the progression of the COVID-19 pandemic. Caution should be exercised when extrapolating these findings.

## 1. Introduction

On 5 May 2023, the World Health Organisation (WHO) calmly declared an end to the COVID-19 pandemic [1]. This was in stark contrast to the frantic announcement at the beginning of the pandemic more than three years earlier (March 2020), when governments around the world scrambled to introduce measures that they thought might control the progression of the pandemic, described as non-pharmaceutical interventions (NPIs). These measures included travel restrictions, stay-at-home orders, mask mandates, school closures and social distancing [2,3] and are adequately described by Hale et al., 2021 [4]. Later, it was believed that the introduction of vaccines in early 2021 would augment these measures, especially by prioritising older age groups, most at risk of severe COVID-19, and front-line workers. These were widely seen as being very effective at reducing the incidence of symptomatic COVID-19 [5,6,7,8,9]. There was also speculation that, if the vaccines could reduce the chances of infection, they could also reduce transmission rates [10], and so governments expanded their vaccination programmes population-wide [11].

COVID-19 has now spread to almost every country in the world, with the highest number of deaths being documented in the USA, followed by Brazil and India [12]. Not only did it have significant consequences for global public-health expenditure and the provision of health services but it also had severe impacts on the socio-economic life of many countries [13,14,15]. It is worth noting that this spread and its repercussions were not uniform and had many geographical, temporal and demographic variations. They were also heavily influenced by the nature of the RNA virus itself, which was characterised by a successive series of prevalent mutations, each having different levels of disease severity and communicability [14,16].

Many studies of the pandemic have argued since then, that its progression and dynamics over time (including the rises and falls of different “waves”) were heavily influenced by government interventions—both NPIs [2,3,17,18,19,20,21,22] and vaccination programmes [11,23,24]. Other studies acknowledged that the dynamics of the pandemic were also substantially influenced by the evolution of different variants throughout the pandemic [25,26] and that there was a corresponding waning in the effectiveness of the original vaccines over time [27,28,29]. However, it was also recognised that specific government NPIs might have coincided with voluntary societal changes made by members of the public without necessarily being explicitly mandated, e.g., social distancing [21]. Nonetheless, an implicit assumption of these studies is that the primary drivers of the dynamics of the pandemic—other than the evolution of variants [25,26]—were the various NPIs and pharmaceutical interventions implemented by governments.

However, there are reasons to suspect that the dynamics of the pandemic were not determined exclusively or even mostly by these factors. First, many of the studies that identified a strong influence of different NPIs on the dynamics of the pandemic implicitly or explicitly assumed (for simplicity) that the NPIs were the primary drivers of the pandemic dynamics [2,3,18,19,20,21,22]. Hence, if a wave began to fall, it was assumed that this was probably a result of changes in NPIs that occurred around this time—rather than the possibility that other factors were involved. In contrast, many studies that did not make this assumption failed to identify clear and substantial evidence of their efficacy [30,31,32,33,34,35,36,37,38,39]. Second, for simplicity, many of the studies suggesting that the NPIs strongly influenced the pandemic dynamics confined their analysis to either the first wave [2,3,18,19,20,21] or the second wave [40]. Therefore, these studies typically did not evaluate the inter-wave dynamics or consider the dynamics over multiple years. Third, while the clinical trials initially suggested that the various COVID-19 vaccines used were effective at reducing the incidence of symptomatic COVID-19 by as much as 95% in the case of Pfizer/BioNTech [5] and by 76% in the case of Astra Zeneca [7], these estimates were based on relative risk reductions (RRR) involving a relatively small number of COVID-19 cases across either arm of the vaccine trials [41,42,43]. For example, while the Pfizer/BioNTech (New York, New York, USA) clinical trial involved 43,548 participants divided roughly equally between the vaccinated and placebo arms of the trial, the total number of confirmed symptomatic COVID-19 cases identified across both arms of the trial was only 170 [5]. Therefore, while 95% of those cases were in the placebo arm, yielding a statistically significant relative risk reduction (RRR) of 95%, the statistical sample used for evaluating vaccine efficacy was very modest [41,42,43]. We recognise that multiple studies have reported evidence that the vaccination programmes were a major influence on the pandemic dynamics from 2021 onwards [11,23,24], which initially seems to support those promising clinical trial results, but other studies suggest that the effectiveness of the vaccines on the pandemic dynamics were much less than expected [27,43,44].

Finally, other human coronaviruses in the same family as SARS-CoV-2 (i.e., the coronavirus responsible for COVID-19 illness) exhibit a strong seasonal trend in their incidence—peaking in mid-winter and having a very low incidence during the summer [45,46,47,48,49]. Some studies in the early stages of the pandemic suggested that this seasonality in coronavirus incidence might also influence the dynamics of the pandemic [45,46,47,48,49,50,51,52,53]. Others disagreed with this assessment and argued that COVID-19 should not be treated like other “seasonal” infections because seasonality could not be (solely) used to characterise the early pandemic progression, e.g., outbreaks during summer and spring periods [54,55,56,57,58]. Nonetheless, while these points rule out seasonality as the sole factor, most mid- to high-latitude countries experienced major COVID-19 waves during the winter months of 2021–2022, and marked reductions during the summer months, suggesting that seasonality is at least a contributing factor [50,52,59,60,61,62]. Surprisingly, most of the standard mathematical models used for modelling epidemics that inform governments on the appropriate and/or required responses to the pandemic did not consider the role of seasonality [63].

Therefore, this study examines the seasonality and influence of public-health interventions on the COVID-19 pandemic in Northern Europe. We hypothesise that much of the apparent success of NPIs and vaccination in bringing the pandemic under control might actually have been an effect of seasonality. We chose pharmaceutical (primarily vaccination) and non-pharmaceutical interventions because they collectively represent the bulk of governmental and societal efforts to alter the progression of the pandemic. However, unlike earlier studies, our analysis uses empirical data and does not make any assumptions as to the effectiveness or otherwise of government interventions. We have also made generous allowances, through time-lagged analysis, to detect any possible influences of these policies on the progression of the pandemic.

If the pandemic had a significant seasonal component, then we would expect the pre-pandemic seasonality of other beta-coronaviruses to serve as a reasonable approximation of the expected seasonal component of the progression of the pandemic, independent of interventions. Therefore, the influence of seasonality is probably best observed in northern latitudes, where seasonal peaks of endemic beta-coronavirus are perhaps more pronounced [49], even though studies from more tropical climates such as Nigeria, Democratic Republic of Congo (DRC), Senegal, and Uganda also indicate, to some extent, the seasonal nature of COVID-19 [64,65]. Europe is perhaps one of the most interesting areas to study in this respect because, although it only has 10% of the world’s population, it had, by the end of 2022, the highest incidence of cases worldwide [12] accounting for nearly 36.8% of all cases and 29.5% of deaths [12]. However, even in Europe, there are still differences between the four regions, North, South, East and Western Europe [12]. For this study, we chose Northern Europe because of the consistency of the epidemiological data on COVID-19, the good access and availability of healthcare services, and the availability of a dataset for the long-term incidence of other endemic coronaviruses in one of the countries in the region (Sweden).

Even though there have been many studies of the effects of NPIs [17,66,67], vaccinations [11,44], seasonality [52,60,62,68] and limited combinations of these [61] on the progression of the COVID-19 pandemic, at local and international levels, there are very few that evaluate all three potentially significant influences in a Northern European context. Additionally, previous studies of the possibility of the seasonal influences of COVID-19 have been uncertain, due to limited coverage. To address such gaps in our knowledge and their associated uncertainties, we considered it appropriate to evaluate the progression of the whole officially documented pandemic for six neighbouring Northern European countries, (Ireland, the UK, Sweden, Denmark, Norway and Finland from 1 March 2020 to 6 May 2023) in terms of three factors: (1) the stringency of NPIs; (2) the vaccination programmes; and (3) the seasonality of human coronaviruses.

The results of our studies here could help public-health decision-makers prepare for similar outbreaks of COVID-19 and inform them about the types of policies that were effective in controlling its progression. However, caution should be exercised when extrapolating these results to other areas with different climates and socio-economic statuses. In particular, we caution that our analysis is a population-based observational ecological study evaluating the progression of the pandemic at a population level only as distinct from an experimental-based study or an individual-based observational study.

## 2. Methodology

For our analysis, we carried out a population-based observational ecological study of six countries in Northern Europe. To do this, we statistically compared these three factors over time to the pandemic dynamics (including timings and magnitudes of the rises and falls of each “wave”) for each of six neighbouring Northern European countries (Ireland, the UK, Denmark, Sweden, Norway and Finland) with an estimated population of 99.8 million in 2020. We first analysed the pandemic dynamics in terms of each of the three factors individually, i.e., NPIs, deaths and seasonality, using Pearson correlation coefficients and time-lagged analysis—to consider their contributions in isolation. To evaluate how the three factors complemented each other over the course of the pandemic, we used multivariable statistical analysis. This involved modelling all three (potentially influential) factors at the same time, i.e., NPIs, vaccinations and seasonality, against the progression of the pandemic as measured by deaths or cases. Checking that all three factors produced a reasonable simulation of actual progression of the pandemic, we removed one of these factors at a time, i.e., NPIs, vaccinations or seasonality, and assessed how much of the simulated progression curve the particular variable was responsible for. If this difference was significant, then this factor was probably responsible for a lot of the variation in the progression of the pandemic.

For the data on the progression of the pandemic as well as the NPI stringency and vaccination rates, we used data from the “Our World in Data” website https://ourworldindata.org/COVID-cases (accessed on 25 July 2023) [69]. This was one of the first databases in the world dedicated to the compilation of COVID-19 data and has been proven to be a reliable source of data. It is freely available, easy to download, and routinely cited in many studies [69].

As a proxy for the seasonality of human coronaviruses, we availed a useful dataset generated by Neher et al. (2020) [45]. This dataset compiled the relative incidence of the four endemic human coronaviruses recorded over a ten-year period immediately prior to the start of COVID-19 (1 January 2010 to 2 April 2020) in Stockholm, Sweden, and therefore was immediately relevant to the geographical area and to current trends in endemic beta-coronaviruses [45]. These four coronaviruses (two alpha-coronaviruses and two beta-coronaviruses) have been suggested to be responsible for 10–15% of “common cold” cases and demonstrate a strong seasonality. Given that COVID-19 is caused by a human beta-coronavirus, we suggest that, if it also has a seasonal component, it might be similar to the seasonality of these endemic beta-coronaviruses (HCoV-OC43 and HCoV-HKU1). Since our seasonality proxy is based on data collected in Sweden, we have confined our analysis to Sweden and five countries in geographical proximity. Indeed, the six Northern European countries (Ireland, the UK, Denmark, Sweden, Norway and Finland) chosen in this study are defined as a biogeographical region in the biogeographical system (World Geographical Scheme) for Recording Plant Distributions [70]. (We note that future research could expand on our analysis by using similar data from other geographical areas, e.g., the United States [46].)

### 2.1. Data

Daily case numbers, vaccination rates, positive test rates, hospitalisation, ICU occupancy and stringency of NPIs covering the period 1 March 2020 to 6 May 2023 were downloaded from the “Our World in Data” website (https://ourworldindata.org/COVID-cases [69]; accessed on 25 July 2023) for Ireland, the UK, Denmark, Sweden, Norway and Finland, which are collectively described as the Northern European countries.

Daily case, death and test data were converted into the equivalent weekly time series by summing the totals for each week. For vaccination rates, the number of fully vaccinated individuals was expressed as a weekly maximum percentage of that country’s population. For the NPI stringency index, the 7-day median value was used to represent the weekly value. Definitions of all these metrics can be found in Appendix B.

Weekly beta-coronavirus (HCoV-OC43 and HCoV-HKU1) cases (1 January 2010 to 2 April 2020) were recorded at the University Hospital in Stockholm, Sweden. These data were used as a proxy for the typical human coronavirus seasonal signal across all six Northern European countries (Figure 1a) [45]. The methods used to collect the prior Swedish data are provided in detail in the article “Potential impact of seasonal forcing on a SARS-CoV-2 pandemic” [45]. Weekly beta-coronavirus cases were expressed as a percentage of the total number of identified cases for that year. The “year” was defined as centred on the peak infection period during the winter and spanned from epidemiological week 26 through to the same week of the next year (Figure 1b), rather than the calendar year. These percentages were averaged for each epidemiological week over a 10-year period (Figure 1c). For simplicity, the analysis in this manuscript is based on this 10-year average (Figure 1c), however, as seen in Figure 1b, the exact timings and magnitudes of the peaks vary slightly from year to year. Future studies might try to account for this interannual variability.

### 2.2. Statistical Analysis

GraphPad Prism (version 6.07) was used to compare the time series (1 March 2020 to 6 May 2023) for the progression of the COVID-19 pandemic as measured (deaths, cases, hospitalisations, ICU occupancy and the positivity rate) for each Northern European country (“Our World in Data” https://ourworldindata.org/coronavirus; accessed on 25 July 2023 [69]) for the three potential driving factors for the progression of the COVID-19 pandemic under consideration (i.e., NPIs, vaccinations and seasonal incidence of beta-coronaviruses) using Pearson correlation coefficients, commonly known as *r* values.

The Pearson coefficient is a simple, standard measure of linear correlation between two paired variables (measured on an interval or ratio scale) in a dataset that ideally requires the variables to be approximately normally distributed without extreme outliers. The data used in our calculation of the Pearson correlation coefficient are representative, paired, weekly samples with no sudden fluctuations from which a linear relationship between the two variables is reasonably expected.

This coefficient can be any value from *r* = 1.0 (perfectly correlated) to *r* = 0.0 (zero correlation) to *r* = −1.0 (perfectly anticorrelated). We evaluate the statistical significance of the *r* values using the associated *p* values, where we use the common (although somewhat arbitrary) threshold of *p* < 0.05 to define “statistically significant”.

In some instances, we also considered the effects of lagging the different time series relative to each other by up to 8 weeks on the correlation coefficients.

Afterwards, multivariable regression analysis was used to consider the relationship between the three proposed factors (NPIs, vaccinations and the seasonality of beta-coronaviruses) and the progression of the pandemic. All three factors, i.e., NPIs, vaccinations and seasonality, were simultaneously modelled against the progression of the pandemic. Each influential variable was then removed one at a time to assess how much this variable contributed to the overall dynamic of the progression of the pandemic. If this difference was substantial, then it was assumed that this factor significantly affected the dynamics of the pandemic.

### 2.3. Study Area

Our study area is the six neighbouring Northern European countries (Ireland, the UK, Denmark, Sweden, Norway and Finland) as defined by a biogeographical system (World Geographical Scheme) for Recording Plant Distributions [70]. These countries were chosen (a) because all six countries have relatively similar demographics, GDPs, etc., as developed Northern European countries and (b) due to their close geographical proximity to the source for our seasonality proxy, i.e., the incidence of beta-coronaviruses in Stockholm over a ten-year period [45]. Small islands in the same biogeographical group and Iceland were excluded because the population sizes were too low for our analysis (Figure 1).

### 2.4. Describing the Progression of the COVID-19 Pandemic for Each Country

Case numbers have been widely used by many countries to describe the progression of the COVID-19 pandemic and allow timely comparisons between studies. However, changes in case definitions, testing capacity and testing priorities over the course of the pandemic have made this metric quite inconsistent for accurately assessing its progression [21,22,71,72]. In the early stages of the pandemic, as testing capacity was being developed and scaled up, tests were typically prioritised for those exhibiting the most severe symptoms and/or front-line workers [73,74]. As a result of this, case definitions were often quite restrictive. However, as testing capacity increased and the first wave of COVID-19 started to decline, increased supply and reduced demand allowed laboratories to develop less restrictive case definitions and testing priorities. As a result, the true number of infections during the first wave of the pandemic was probably substantially underestimated [75,76,77,78].

After the first wave (corresponding to summer 2020 in the Northern Hemisphere), many countries (including the Northern European study region) had significantly increased their testing capacity to a point where all potential cases, irrespective of symptoms, could be tested. Moreover, given that the viral load appeared to increase rapidly in a 1–2 day period before symptoms became apparent [79], many laboratories were encouraged to conduct RT-PCR tests up to a very high cycle threshold (Ct) to reduce the incidence of false negatives during the pre-symptomatic period [80]. Although this trend reduced the chances that pre-symptomatic individuals might mistakenly avoid quarantine due to a negative result, it also meant that many of these later “identified cases” were neither infectious nor symptomatic [81,82]. Accordingly, the incidence of “identified cases” since mid-2020 has partially been a function of testing capacity and testing policies—with testing capacity generally increasing month on month. Public figures who have noted this point have been criticised for not appropriately communicating that the number of tests carried out is also a function of demand as well as supply [83,84,85]. Nonetheless, this makes case numbers a poor metric for studying the progression of the pandemic.

On the other hand, there are also problems with the other widely used metrics for evaluating the progression of the pandemic. These other metrics include deaths from/with COVID-19 [2,22,37], positivity rate [83,84,85], weekly mortality data [38], hospitalisations and ICU occupancy. (Note that using deaths as a metric introduces an additional complexity of a time lag, given that when death occurs it is typically several weeks after infection. Although the exact length of time from infection to death varies between individuals and even its average value is still uncertain, several analyses estimate it to be approximately 3 weeks [2,37,86].)

To illustrate the challenges in using each of these different metrics for evaluating the progression of the pandemic, let us compare these metrics for Denmark, which lies halfway across the geographical range of Northern European countries (Figure 2).

If only weekly case numbers (Figure 2a) are considered as a measure of the progression of the pandemic, it would create the impression that most of the pandemic occurred in late 2021/early 2022. However, as shown via corresponding figures for deaths (Figure 2b), there were in fact three waves during this period.

The time lag between the peak in cases and deaths in successive waves of the pandemic can be used to approximate the interval between infection and fatality. For Denmark, we estimated the time lag between infection and death was between 0–1 week for the first wave, 3–4 weeks for the second wave and 4–5 weeks for the third wave (Figure 2g). We repeated these estimates for the other 5 Northern European countries in this study, which all show a similar pattern (Appendix A).

Our estimates reveal that the time lag between infection and death for the first pandemic wave in March/April 2020 was the shortest (Appendix A). This could be due to the fact that testing was often prioritised for the more seriously ill during the first wave. These cases would typically have been identified several weeks after infection. The time lag period for the second wave was generally longer, perhaps because the increase in testing capacity and less restrictive case definitions and/or testing priorities allowed cases to be identified much closer to the time of infection [87]. The time lag for the third wave in many cases was longer again still [72].

Changes in the testing positivity rate could also be used as a metric for measuring the progression of the COVID-19 pandemic (Figure 2c). This partially accounts for the long-term changes in supply of tests, since it records the fraction of all tests carried out that are positive. This means that it is independent of the number of tests carried out [83,84,85]. This metric also reveals that there were three major waves during the pandemic but suggests that winter 2020/2021 was relatively modest compared to the other two waves.

Although data for hospitalisation (Figure 2e) initially looks like a more promising metric, the length of these records is not consistent across five other Northern European countries. In some countries (Ireland, the UK, Finland and Norway) these records stop before the end of 2022.

This is also similar to the situation for ICU occupancy (Figure 2f) where record keeping was terminated for a number of countries sometime in 2022.

In terms of positivity rates (Figure 2c), by late 2021/early 2022, many countries had introduced “rapid antigen tests” for COVID-19 that could be carried out at home. Although less sensitive than the RT-PCR tests, these offered the general population a relatively easy and quick self-test [88]. Once this option was introduced, it could have significantly altered the positivity rates in different ways depending on how the self-reported antigen test results were reported by each country [88]. E.g., if a country treated a positive antigen test as equivalent to a positive RT-PCR test, then the self-reporting of positive antigen tests could increase the positivity rate. Similarly, if confirmation RT-PCR tests were necessary, then the preliminary antigen test results could introduce a self-screening effect, increasing the positivity rates.

In terms of deaths, we need to recall that in many cases a “COVID-19 death” was recorded where the deceased tested positive for SARS-CoV-2 but where the death was not necessarily caused by the virus. Therefore, it is still unclear what fraction of the reported “COVID-19 deaths” were (a) deaths caused by a COVID-19 infection (“deaths from COVID-19”); (b) deaths where COVID-19 was a contributing factor; or (c) deaths unrelated to COVID-19 but where the person also had COVID-19 (“deaths with SARS-CoV-2”) [75,89,90,91].

Meanwhile, in terms of hospitalisation and ICU statistics, as testing capacity increased during the course of the pandemic, it became increasingly standard for healthcare facilities to routinely test all patients for COVID-19 to ensure that they were treated in separate wards to reduce nosocomial infections [73,92]. An inadvertent consequence of this policy might have been an increase in the fraction of hospitalisations “with COVID-19” where the hospitalisation was unrelated to COVID-19. Indeed, Vu et al. (2022) estimated that only 32.5% of the SARS-CoV-2 positive hospitalised patients in a particular hospital during the period December 2021 to January 2022 (during the local Omicron variant surge) were hospitalised due to COVID-19 [92].

It should be apparent that all of the commonly used metrics for the progression of the pandemic have their own problems [75]. Hence, we repeat our analysis using each of the above metrics in turn. However, for brevity and to reduce repetition, in the main manuscript, we mostly confine our analysis to the time-lagged death (death minus three weeks) as our main metric for evaluating the progression of the pandemic (Figure 2d) [2,37,38]. However, the equivalent analyses in terms of positivity and case numbers, hospitalisation and ICU occupancy can be found in the Appendix A. We have also used case data as a comparison when analysing correlation coefficients because it is one of the most frequent types of surveillance data used during the pandemic [22]. Additionally, we have also included a figure of the progression of the main variants of concern from the beginning of the pandemic in supplementary section (Appendix A).

## 3. Results and Discussion

### 3.1. Influence of Non-Pharmaceutical Interventions on Pandemic Progression in Northern Europe

The introduction of NPIs came shortly after the declaration of a pandemic by the WHO in most Northern European Countries [93]. Many governments and some of the public initially anticipated that these measures would act like a quick circuit breaker to “flatten” the approaching pandemic curve [17,91]. Some of these measures included lockdowns, travel restrictions and school closures [17,37], sometimes backed by strict enforcement of fines and legal action for non-compliance. Whenever policy-makers from one administrative region increased the stringency of NPIs, others from neighbouring administrations seemed to follow suit [94], with some notable exceptions such as Sweden, which generally kept to its original pandemic preparedness plans [91].

Figure 3 compares the weekly stringency index (0–100, where 100 represents the most stringent restrictions) for each country to the corresponding weekly recorded COVID-19 deaths (time-lagged by 3 weeks). The equivalent plots for cases (Appendix A), positivity rates (Appendix A), hospitalisations (Appendix A) and ICU occupancy (Appendix A) can be viewed in the Appendix A. Details of the original data for these plots and subsequent correlations can be found in Appendix A.

Although some authors have pointed out that there can be differences in the type and order of NPIs implemented by each country [95], the overall trend in our data is broadly similar, insofar as all six countries dramatically increased the stringency of NPIs during the early phases of the pandemic (spring 2020)—including Sweden, albeit their NPIs were publicly criticised for a lack of stringency [91,96]. All countries partially reduced NPI stringency during the summer of 2020, followed by an increase in autumn/winter 2020 and a subsequent reduction during spring/summer 2021. Significant increases in the stringency of NPIs were implemented during autumn/winter 2021 in concurrence with the arrival of a new variant of the virus, dubbed “omicron” [25,26]; before reducing the stringency during the spring of 2022 (Figure 3).

In theory, if the NPIs were as effective at reducing the spread of the virus as expected, the progression of the pandemic should be negatively correlated with the stringency index. That is, as NPI stringency increases, the number of deaths/cases should decrease at some time after their introduction and vice versa. We might also expect the response of the pandemic to the NPIs to possibly be lagged by a few weeks, given that there is often a latency period between infection and detection. For example, we would expect that after an effective NPI is introduced (increased stringency), there should be a reduction in cases, deaths, positivity rate, hospitalisations and ICU occupancy at some point after this. Conversely, if an effective NPI is removed prematurely (decreased stringency), we might expect to see a resurgence in infections over the coming weeks. However, our time series shows that the peaks and/or plateaus in NPI stringency often follow the waves in the progression of the pandemic, i.e., the opposite of what should be expected.

For instance, in Ireland, after the NPIs were introduced in March 2020, the NPI stringency remained at its highest value for a few months before it was significantly reduced over the summer of 2020. Then, it was steadily ramped up throughout the autumn and winter before slowly being reduced in spring 2021 and substantially throughout summer/autumn 2021. After a brief ramping up during the winter, the NPIs were reduced further during spring 2022 to their minimum since their introduction. Yet, when we consider the progression of the pandemic in terms of (time-lagged) deaths (Figure 3), it can be seen that the changes in NPIs generally followed the progression of the pandemic, rather than the other way around. That is, deaths generally rose during periods when NPIs were increasing and NPIs generally were reduced after the peaks in deaths had already occurred. We can see similar results for Ireland using cases (Appendix A), positivity rates (Appendix A), hospitalisations (Appendix A) and ICU occupancy (Appendix A).

One plausible explanation could be that governments were increasing or decreasing the stringency of NPIs in response to the progression of the pandemic. This was the case, for instance, for Hong Kong’s “suppress and lift” strategy, in which interventions were progressively strengthened whenever the incidence of infections rose, and relaxed whenever they declined [97]. This might make sense politically since policymakers might feel a responsibility to take action, i.e., increase NPI stringency, during periods when the virus seems to be spreading rapidly. Then, when the waves have been steadily falling, policymakers might feel it would be time to temporarily “relax” the restrictions. However, this is the opposite of what should have occurred if the NPIs were very effective, i.e., it would mean that the progression of the pandemic was driving changes in the stringency of NPIs, rather than the other way around.

That said, there are two examples where negative correlations are visually obvious, i.e., Finland and Norway. These two countries had very few deaths (per 100,000) throughout the pandemic compared to the other four countries. However, most of those deaths appear to have occurred towards the end of 2021/early 2022—a period that coincided with a general decrease in NPIs to the lowest point since their introduction. We can see similar results for cases (Appendix A) and positivity rate (Appendix A).

As we will discuss later, this is not encouraging for the effectiveness of the vaccination measures for these countries, since one of the main justifications for decreases in NPIs was the high vaccination rates that had been achieved at that stage. Nonetheless, in terms of NPIs, Norway and Finland initially might appear to offer some support for their effectiveness. Therefore, let us investigate this potential support for the NPIs in more detail. If these NPIs are as effective as hoped for, then we should also expect to see similar effectiveness for the other Northern European countries. Similar trends can be viewed in equivalent plots for cases (Appendix A), positivity rates (Appendix A), hospitalisations (Appendix A) and ICU occupancy (Appendix A) in the Appendix A.

In Figure 4, the Pearson correlations are plotted for each country between NPI stringency and either (a) cases or (b) deaths. We also show the effect of lagging the NPI stringency time series backwards by up to 8 weeks and forwards by up to 4 weeks. Details of these correlations can be found in the Appendix A.

The *x*-axis in each plot indicates the extent of this lag. The zero lag compares the weekly NPI stringency directly to the number of (a) cases and (b) deaths for that week. However, a positive (or negative) lag indicates that the correlation is between the NPI stringency for each week and the cases/deaths stringency from several weeks later (or earlier). That is, the cases/deaths time series was shifted forwards (or backwards) by that number of weeks before the correlation was calculated.

A positive lag indicates that the (a) cases or (b) deaths are being compared with the interventions introduced that many weeks ago. This is a physically plausible scenario since there is typically a latency period between infection and the identification of a case. It is especially plausible for deaths since, as we discussed earlier, when an infection leads to death, there can be a lag of several weeks from infection to death.

A negative lag is physically unrealistic for an effective NPI since it is the opposite of the expected direction of causation. That is, a negative lag indicates that changes in the progression of the pandemic are driving changes in the stringency of NPIs, i.e., the opposite of what should be occurring for an effective NPI. However, this physical unrealism also makes these negative lag correlations particularly useful for interpreting the relevance of the statistics. This is because we can use the correlation values for these “anticipatory” lags as a baseline for evaluating how effective the NPIs were for each country in terms of changes in (a) cases and (b) deaths.

In terms of deaths, we can see that for half of the countries (Ireland, the UK and Sweden), the NPI stringency was positively correlated with death for all lags up to 5–8 weeks. And for those longer lags, the slight negative correlations are not statistically significant. This is the opposite of what should be expected if the NPIs were effective at reducing the number of deaths. Instead, it suggests that the NPI stringency was being altered in response to the progression of the pandemic, i.e., the pandemic progression was driving policy changes in NPI stringency, rather than the other way around—as was the case for Hong Kong, for example [97].

On the other hand, for Denmark, Finland and Norway, the deaths remained negatively correlated with NPI stringency for all lags. Initially, this might seem encouraging. However, we note that for all three of these countries, much of the negative correlation already exists for the physically unrealistic negative lags. This suggests that the negative correlations are not causal ones (in the sense of NPIs influencing the course of the pandemic). If they were, then the negative correlations should only be occurring for the positive lags.

In terms of cases, the results might initially seem more encouraging in that all countries have negative correlations as should be expected if the NPIs were effective. However, again, a problem arises in that these negative correlations also exist for the negative (“anticipatory”) lag values. Indeed, for most of the countries (apart from Sweden), the strongest negative correlations occur for a negative lag of −4 weeks, i.e., indicating that the changes in cases were leading to changes in NPI stringency four weeks later. These trends gradually become less negative with more positive time lags, suggesting little or no beneficial effects from NPIs. These observations are consistent with a study by Bjørnskov, which used weekly mortality data instead of COVID-19 deaths as a metric and found that the estimated effects of lockdown policies are all positive and significant when policy changes are lagged by one or two weeks [38]. However, when the lag length is extended to three or four weeks, i.e., the length that is reasonable from the perspective of the virology of SARS-CoV-2, the estimates become very small and insignificant [38].

In conclusion, we cannot detect any clear evidence in our analysis to suggest that NPIs have significantly altered the course of the pandemic for these six Northern European countries in our study. However, we stress that the absence of evidence is not necessarily evidence of absence, and that correlations do not necessarily mean causations. It is plausible that changes in the NPI stringency had effects on the progression of the pandemic, but that these effects were quite modest—and thereby too small to be detected by our method of analysis. Moreover, since we are dealing with population-scale statistics, we should be wary of the so-called “ecological fallacy” whereby population-averaged statistics might be (incorrectly) assumed to apply equally to all individuals in that population. It may well be that, for example, NPIs helped some individuals reduce their short-term chances of exposure to potential infection more than others.

Nonetheless, if the NPIs are as effective as commonly assumed, we would expect to see some clear, consistent and/or unambiguous evidence for this. On the contrary, we do not find any evidence that the NPIs had a significant influence on the pandemic dynamics. We acknowledge that this differs from the conclusions of several previous studies, which argue that the NPIs were effective in altering the course of the pandemic, e.g., [2,3,18,19,20,21,22]. However, it is consistent with the conclusions of other studies, e.g., [30,31,32,33,34,35,36,37,38,39,63,91,96]. In particular, a similar analysis by Mader et al. using the same data source (Our World in Data; 1 July 2020 to 1 September 2021) did not find any substantial and consistent COVID-19-related fatality-reducing effect from the NPIs [33]. Wood, made a similar point quite early in the pandemic, that although cases or deaths were in decline this was not necessarily the product of NPIs; indeed, he noted that cases began to go into decline before government lockdowns were introduced in the UK [37]. Other researchers such as Bjørnskov, who studied 24 European countries, state that they could find no clear association between lockdown policies and mortality development [38].

Of course, we acknowledge the intuitive expectation that many of the NPIs should significantly alter the progression of the pandemic. Hence, many studies assumed that the various NPIs must be working to some extent and focused on quantifying the relative effectiveness of individual measures [98,99]. However, we think by modelling time-limited datasets and not factoring in the potential influences of other confounding variables such as seasonality, earlier researchers might have mistakenly attributed changes in the progression of the pandemic to NPIs. We stress that scientific insights are often non-intuitive and if intuition were a reliable source of knowledge, scientific observation would be largely unnecessary. Therefore, we recommend that assessments of the effectiveness of NPIs should be based on a critical evaluation of the data, rather than our expectations of what should be.

### 3.2. Influence of Vaccinations on Pandemic Progression in Northern Europe

Figure 5 compares the weekly cumulative percentage of the population vaccinated for each Northern European country to the corresponding weekly data on deaths (time-lagged by 3 weeks). The equivalent plots in terms of cases (Appendix A), positivity rates (Appendix A), hospitalisation (Appendix A) and ICU occupancy (Appendix A), are available in the Appendix A. Details of the data and subsequent correlations can be found in Appendix A.

Vaccination programmes began in earnest in the Northern European countries at the beginning of 2021 (Figure 5). As mentioned in the introduction, these vaccines had a very high reported relative risk reduction (RRR) for symptomatic COVID-19 [5,6,7,8]. Therefore, if the vaccines were as effective as suggested by those clinical trials, the incidence of symptomatic COVID-19 and thereby the average transmission rate should have tended to decrease as the vaccinated percentage of the population increased over time. The rationale is that “uninfected people cannot transmit; therefore, the vaccines are also effective at preventing transmission” [10]. Additionally, if the vaccines were as effective as the clinical trials suggested, then there should be a general anticorrelation with the progression of the pandemic in terms of cases and positivity rate. Although the vaccination campaign began in January 2021, the vaccines were initially reserved primarily for the elderly and front-line workers. Once these groups had been offered the opportunity, vaccination was sequentially expanded to successively younger generations, so that by mid-2021, the opportunity to receive the vaccine was available for all adults. This was maybe in contrast to other lower-income European countries that were not as fast with vaccine roll-out [100] or even other medium- to low-income nations where the availability of vaccines was limited [101,102]. Vaccination programs targeting children and young adults did not begin until late 2021.

The number of deaths relative to the total population and infection number decreased shortly after the introduction of vaccines, continuing into the spring and summer of the same year, prompting many to infer that the vaccination programmes were successfully beginning to end the pandemic [23,24]. However, as shown (Figure 5), throughout autumn/winter 2021, deaths began increasing again despite the percentage vaccinated (all ages) being on average of 72.6% (min 70.74%–max 77.05%).

Several explanations have been offered for this—chiefly suggesting some combination of the evolution of new variants of the virus and/or the possibility that the vaccine efficiency wanes over time [28,29]. However, we note that there are reasons to consider the possibility that the vaccines were simply not as effective as originally hoped. For instance, studies from June–August 2021 found that the mean viral loads were similar for vaccinated and unvaccinated individuals infected with SARS-CoV-2 during the Delta variant surge regardless of symptoms [103]. Additionally, a UK Health Security Agency report (3 March 2022) found that the rates of all COVID-19 cases were between 1.7 (80 years or over) and 3.4 times higher (40–49 years) among those who had received at least three vaccine doses compared to the unvaccinated in all age groups of 18 years and older [104]. This suggests that the promising initial claims that these COVID-19 vaccines were very effective at reducing the likelihood of infection [5,6,7,8] were not as robust as hoped. Indeed, Kampf (2021) noted that the high “rate of symptomatic COVID-19 cases among the fully vaccinated breakthrough infections” since July 2021 contradicts the expected reduction in transmission among the vaccinated population [27].

Despite the high incidence of “breakthrough infections”, some justified the continued use of COVID-19 vaccines as a means of substantially reducing COVID-19 severity and/or death [28,29,105]. However, although the magnitude of the third pandemic wave seemed to be reduced for Ireland, the UK and Sweden after the introduction of vaccination (in comparison to the first two waves), the opposite was observed for Demark, Finland and Norway, i.e., these countries had a comparatively larger third wave than the preceding two waves (Figure 5). These unexpected trends are even more pronounced if the progression of the pandemic is measured through cases (Appendix A), positivity rate (Appendix A), hospitalisations (Appendix A) or ICU occupancy (Appendix A). Therefore, as for NPIs, we should be careful not to prejudice our analysis of the effectiveness of these COVID-19 vaccines with our expectations of what should be.

The statistical correlations (positive or negative) between the vaccination and the progression of the pandemic are formally considered in Figure 6. Details of the correlations can be found in Appendix A. The Pearson correlations between the percentage of the population fully vaccinated and (a) cases or (b) deaths are plotted for each country. As before, we also show the effect of lagging the vaccination rate time series by up to 8 weeks, i.e., what would happen if the effects of vaccination were not apparent for up to 8 weeks. We also repeated this process in the other direction by 4 weeks, which would be physically improbable, but it would be useful to detect any underlying trends in the data before vaccines were administered.

Again, the *x*-axis in each plot indicates the extent of this lag. A positive lag indicates that the (a) cases or (b) deaths are being compared with the percentage of people vaccinated several weeks ago. We would expect the reduction in cases or deaths to begin after an effective vaccination programme was introduced. As discussed earlier, there may also be a lag of several weeks before infection data translates into deaths.

There should also be a time-lag between the administration of the vaccines and the development of effective immunity. The vaccine manufacturers suggest that this lag is two weeks after the final dose and these lags are already incorporated into the definition of when a person is “fully vaccinated”. However, it is plausible that the exact lags vary between individuals.

A zero lag in this case compares the weekly percentage of the population vaccinated directly to the numbers of (a) cases and (b) deaths for that week.

Again, in Figure 6, a negative time lag is physically unrealistic since it inverts the apparent direction of causation, i.e., it would indicate that the progression of the pandemic is causing changes in the level of the population vaccinated, rather than the other way around. However, again, these negative lag correlations are a useful baseline to evaluate the effectiveness of the vaccination programme in terms of (a) cases and (b) deaths.

In terms of cases, we would expect to see a negative correlation between the vaccination rates and the progression of the pandemic. That is, as the vaccination rates increased, we would expect the incidence of cases to generally decline, perhaps with a lag of a few weeks. However, for many of the countries, there is a positive correlation (Finland, Norway, Denmark and Ireland) which seems perfectly level through all time lags except in the case of Norway where there is a barely perceptible increase in correlation. The correlation values for Sweden remain negative but not significant and the values for the UK slowly move from a negative to a positive correlation although all these values are also not significant (Figure 6a).

Therefore, this analysis failed to identify any evidence that the vaccines reduced the incidence of cases in any of the Northern European countries. This is surprising given that the relative risk reduction (RRR) for symptomatic COVID-19 was relatively high—up to 95% for the Pfizer vaccine [5]. However, as others have noted, the absolute risk reduction (ARR) observed during these trials was very modest ranging from 0.71 % for the Pfizer vaccine to 1.22% for the Astra Zeneca vaccine [41,42]. This is because the total number of confirmed symptomatic COVID-19 cases identified in either arm of the trials was very modest—typically about 200 cases per vaccine trial [5,6,7,8]. Hence, while the trials had fairly large participant samples, e.g., 43,548 participants for the Pfizer trial [5], and most of the identified cases were in the placebo/control group—yielding a high “RRR”, these initially promising RRR values might not have been as robust as first hoped.

That said, although most of the clinical trials did not have enough data to estimate whether there was a statistically significant effect on death, some of the studies qualitatively suggested that the vaccine might reduce the incidence of death associated with COVID-19 [7]. So, let us now consider the influence of vaccination rates on the pandemic progression in terms of deaths (Figure 6b).

When we consider the zero-lag correlations, the results initially appear more encouraging in that deaths are negatively correlated with vaccination rates for half of the studied countries (Ireland, the UK and Sweden) although the negative correlation for Sweden is not statistically significant. This initially promising result becomes less encouraging, however, once we consider how the correlations vary with the lag. For these three countries, the most negative correlations occur for negative (“anticipatory”) lags. Yet, these negative lags are physically unrealistic for the expected effect. If the vaccination rates were strongly reducing the numbers of deaths in these countries, then the negative correlations should be strongest for the positive lags, rather than the negative lags.

Meanwhile, for the other three countries (Denmark, Norway and Finland), we can see that vaccination is positively correlated with death for all lags up to 8 weeks. Again, this is the opposite of what should be expected if the vaccination programme had been effective in reducing the number of deaths.

Therefore, our analysis also fails to identify any evidence that the vaccines have reduced the number of COVID-19 deaths in any of the Northern European countries. However, this does not exclude the possibility that further studies might find such evidence. Additionally, we remind the reader that our observations apply to the national total statistics for the countries. Therefore, we should be aware of the ecological fallacy and recall that these findings referring to the overall net pandemic progression for each country might potentially mask potential benefits for some individuals that are not apparent at a country-wide scale [106]. Nonetheless, again, if these COVID-19 vaccines were as effective as the original clinical trials implied [5,6,7,8,9], then we would expect to see a clear signal of this from our analysis.

In contrast to our findings, Watson et al. (2022) claimed to have found a strong global reduction in cases and deaths as a result of the vaccination programme [11]. However, this analysis was based on a comparison of the observed data to a counterfactual model scenario. Essentially, they concluded the vaccines must have been very effective because their model predictions of what should have occurred in the absence of vaccination programmes failed to transpire. However, it might be that these studies did not account for other confounding factors as a probable cause for changes in the progression of the pandemic and therefore the decrease in either deaths or cases was attributed to vaccinations.

That said, we have also seen many non-model-based studies reporting that the vaccines have been effective [9,23,24,107,108], albeit some argue that this effectiveness wanes with time and/or new variants [28,29,109,110]. However, we note that (as for NPIs), these studies typically assume from the outset that the vaccines are very effective and that any changes in the progression of the pandemic must be related to some combination of the vaccines and/or the NPIs [23,24,28,29,105,107,108,109,110]. Therefore, they typically overlook the possibility that there could be other confounding factors, e.g., coronavirus seasonality, that need to be accounted for.

We suggest that rather than assuming these vaccines must be very effective, we should approach the data with an open mind. We note that a global comparative study by Subramanian and Kumar (2021) that appears to have taken this approach concluded that “increases in COVID-19 are unrelated to levels of vaccination across 68 countries and 2947 counties in the United States” [44], as measured by case numbers. This surprising conclusion appears similar to our findings [44]. Although Subramanian and Kumar were criticised for only considering case numbers [105], our analysis considers multiple metrics and still comes up with similar findings at least for the six countries studied here. To strengthen the conclusions from our data, we note that all six Northern European countries had several more pandemic waves (beginning in August 2021) after having reached high levels of vaccination. We emphasise that our analysis is confined to these six Northern European countries, therefore great care should be taken when trying to extrapolate this data to other situations.

### 3.3. Influence of Seasonality on Pandemic Progression in Northern Europe

The seasonal nature of many infectious viruses has been known for centuries to the extent that they have been incorporated into the popular vernacular as “cold and flu season” [47]. Indeed, studies in Northern European countries have identified the seasonal nature of rhinovirus, adenovirus, influenza A and B viruses, human parainfluenza viruses 1–3 (HPIV), respiratory syncytial virus (RSV) and human metapneumovirus (HMPV) concluding that these seasonal variations are associated with specific meteorological factors [111]. Additionally, some other studies point out that changes in human behaviour during peaks in these infections are affected by contact rates between infected and susceptible individuals [47,112]. These factors have further implications for both innate and acquired immunity as well as the stability of viruses such as the beta-coronaviruses [47]. There have also been many studies of the effects of environmental factors on the viability and transmission of COVID-19. Research has shown that the virus can be inactivated by high levels of UV light or sunlight irrespective of levels of interfering substances [113] and that both sunlight and temperature and to a lesser extent humidity influence the persistence of COVID-19 [114]. Other research has shown that solar radiation decreases the transmission of SARS-CoV-2 [115], which has obvious seasonal implications since sunlight is at a maximum during summer months. Recent research has also indicated that higher levels of UV light and mean temperatures may contribute to a reduction in the transmission of COVID-19 as well as a reduction in its incidence, hospitalisations and mortality [116,117].

At any rate, seasonality appears to play an important role in many respiratory illnesses including several other human coronaviruses, and therefore potentially SARS-CoV-2 [45,46,47,48,50,51,52,60,91,118]. Some studies have argued that seasonality was not a major factor in the dynamics of the pandemic [54,55,56]. However, others have argued that it should be an important factor to be considered [50,52,57,58,59,60,61,118]. This lack of consensus may stem from earlier research on the pandemic that relied on differences in geographical locations as a substitute for seasons, in a so-called “space for time” substitution [57] or on limited seasonal data. Now that we have more than three years of empirical data—and a plausible proxy for coronavirus seasonality for Northern Europe—a reassessment of the possible role of seasonality seems appropriate.

Figure 7 compares the decadal-averaged weekly incidence of human beta-coronaviruses (HCoV-OC43 and -HKU1) previously recorded in Stockholm, Sweden (2010–2020) as a percentage of the yearly total with the progression of the COVID-19 pandemic since 2020 in Northern European countries (Figure 7). The equivalent plots in terms of cases (Appendix A), positivity rates (Appendix A), hospitalisations (Appendix A) and ICU occupancy (Appendix A) are provided in the Appendix A. Details of the data and subsequent correlations can be found in Appendix A.

We emphasise that the two sets of plots are derived from completely different time periods and biologically distinct data. The Swedish beta-coronavirus incidences (red dashed lines) are the averages for the 2010–2020 period before the COVID-19 pandemic began and they are the incidences of non-SARS-CoV-2 beta-coronaviruses. Therefore, any similarities between the two plots probably arise from the contribution of the pre-existing seasonality of beta-coronaviruses to the progression of the COVID-19 pandemic.

With that in mind, it is striking that, for all six countries, there appears to be a clear concurrence between the two plots. In particular, winters were accompanied by pronounced waves, and summers were accompanied by very low incidences of COVID-19 (Figure 7). The timings of the winter peaks and summer lows are often strikingly similar to those that would be expected for the endemic beta-coronaviruses (Figure 7). This can also be seen to some extent for cases, positivity rates, hospitalisation and ICU occupancy (Appendix A Appendix A).

That said, we emphasise that seasonality cannot explain all aspects of the pandemic for any of the Northern European countries. More specifically, it is noted that the start of the first wave in March 2020 was “unseasonal” in that it arose several months after the expected winter peak in late December/early January [45,48]. Similarly, the early 2022 peak of deaths for Denmark, Norway and Finland occurred after the expected winter peak. Therefore, some of the early criticism of the possible role of seasonality [54,55,56] might have had some justification.

Nonetheless, unlike for NPIs and vaccination rates, already from this visual comparison, the seasonality of beta-coronaviruses (derived from 10 years of pre-pandemic observations) appears to have strongly influenced the progression of the pandemic.

Now, let us consider the statistical correlations (positive or negative) between the weekly average human beta-coronavirus cases in Stockholm (2010–2020) and the corresponding weekly progression of the pandemic (Figure 8). Details of the correlations can be found in the Appendix A. The Pearson correlations between the weekly average human beta-coronavirus cases in Stockholm (2010–2020) and (a) cases or (b) deaths are plotted for each country. Again, we also show the effect of lagging weekly average human beta-coronavirus cases in Stockholm (2010–2020) time series backwards by up to 8 weeks and forwards by up to 4 weeks.

Again, the *x*-axis in each plot indicates the extent of this lag. A positive lag indicates that the (a) cases or (b) deaths occur after the corresponding week for the average human beta-coronavirus cases in Stockholm (2010–2020). There are two key differences to consider between this natural factor (seasonality) and the previous two societal interventions (NPIs and vaccination rates). First, for the previous factors, we would expect negative correlations for a strong influence. That is, we would expect that as the interventions are implemented, the spread of the virus should decrease. Here, we would expect positive correlations for a strong influence. That is, if SARS-CoV-2 is strongly seasonal like the other beta-coronaviruses, then we would expect the incidence of COVID-19 to rise and fall roughly in time with the 2010–2020 average for the preceding decade.

Second, for the previous factors, a negative lag would be physically unrealistic for an effective intervention, since it would imply that the “effects” of the interventions on the progress of the pandemic occurred several weeks before the interventions were implemented. However, for seasonality, we are comparing the observed COVID-19 trends for 2020–2023 to the 10-year average of the preceding decade (2010–2020). From Figure 1, we can see that there is some variability in the timing of the peaks for each year. Therefore, we should not be surprised if the peaks for one year occur several weeks before or after the decadal average.

In summary, we would expect an increase in cases or deaths to follow rises in the weekly average human beta-coronavirus cases in Stockholm (2010–2020). We would also expect the deaths to lag those of cases by at least several weeks. A zero lag compares the weekly average human beta-coronavirus cases in Stockholm (2010–2020) directly to the numbers of (a) cases and (b) deaths for that week in each of the six Northern European countries. As mentioned above, a negative time lag would mean that the rises in cases and deaths for one of the six Northern European countries would have occurred earlier than that of the decadal average for human beta-coronavirus cases in Stockholm of the preceding decade (2010–2020). However, as explained in Figure 1, even the seasonal peaks for human beta-coronavirus cases in Stockholm are known to have an interannual variation, which can span several weeks (Figure 1b).

One of the first things that we notice about the relationship between the weekly average human beta-coronavirus cases in Stockholm (2010–2020) and the progression of the COVID-19 pandemic as measured by cases is that all the correlations were positive and followed very similar trends (Figure 8a). These positive correlations peaked after 2 weeks for Ireland and the UK, 4 weeks for Sweden and Finland and 6 weeks for Denmark and Norway (Figure 8a). If we look at the corresponding correlations for deaths, we see that for many countries these values are time-lagged by an extra 2 or 3 weeks in comparison to cases except for slight variation with Norway and Finland. For instance, the maximum correlation for Ireland and the UK has lagged by a period of 3 or 4 weeks (Figure 8b).

It is clear from this analysis that there are strong seasonal correlations between the incidence of beta-coronavirus and the progression of the COVID-19 pandemic. We note that the lag with the highest correlation occurred a few weeks earlier for the two British Isles countries (Ireland and the UK) compared to the four Scandinavian countries (Sweden, Finland, Denmark and Norway). One possibility could be climatic differences between the regions. These latitudinal variations have been noted for other seasonal viruses in larger studies of Influenza A and respiratory syncytial virus (RSV) [119]. Perhaps future studies could explore this possibility by comparing the seasonal coronavirus data from other Northern European countries to see how the coronavirus seasons vary between countries.

A similar analysis to our study was conducted by Hoogeveen and Hoogeveen (2021) using the average annual time series for influenza-like illnesses (ILIs) based on incidence data from 2016 to 2019 in the Netherlands [60]. This study found that the time series of COVID-19 and influenza-like illnesses have highly similar seasonal patterns [60]. Another study by Wiemken et al., which covered the USA, Europe and the UK using data from Our World in Data, a study by Fontal et al. in European cities and a study by Townsend et al. of England and Wales came to similar conclusions, i.e., that COVID-19 had a strong seasonal component [52,61,68]. Finally, a study by Shamsa et al., which covered the USA also confirmed that waves of COVID-19 appeared to be seasonal [62]. However, reports of seasonality have not only been confined to northern countries. Studies of the incidence of COVID-19 in the Democratic Republic of Congo, Nigeria, Senegal and Uganda show seasonal trends [64,65]. This perhaps reflects earlier studies of other respiratory viruses, which showed that 50–80% of tropical locations experienced seasonal waves of respiratory viruses [119].

It may be that many of the studies that dismissed the strong influence of seasonality on the progression of the COVID-19 pandemic only had access to time-limited datasets and therefore their modelling assumptions were based on a very limited time period. Additionally, many of these models assumed that changes in the dynamics of the pandemic were from some type of intervention, either NPIs or vaccinations but not from seasonality. In contrast, our studies have the benefit of three years of empirical data that suggest that in the case of these six Northern European countries, the progression of the COVID-19 pandemic is positively correlated with the average seasonality of human beta-coronaviruses (HCoV-OC-43 and HcoV-HKU-1) recorded in Sweden. Additionally, we failed to find a similarly consistent signal for the other two factors we considered (NPIs and vaccination rates) in this geographical area. Therefore, of these three factors, seasonality appears to have been the dominant driver of the COVID-19 pandemic for these six Northern European countries. We have discussed the need for future research to unravel the many complexities and uncertainties in the seasonality of COVID-19, which are discussed in Section 4. Limitations and future research directions.

### 3.4. Influence of Multiple Factors on Pandemic Progression in Northern Europe

In the previous sections, we considered the potential influence of each of the proposed factors on the progression of the pandemic separately, however, these factors are not necessarily independent of each other. For example, governments may have chosen to reduce the stringency of NPIs as the vaccine rate increased under the assumption that the NPIs were no longer as essential. Meanwhile, if there was indeed a strong seasonal component then governments might have adapted their NPIs and vaccination policies in response to the seasonality-driven dynamics of the pandemic progression [97].

To examine the influence of single factors as part of the overall influences on the progression of the pandemic, we used multivariable analysis. This involved modelling the combined (potentially influential) factors at the same time, i.e., NPIs, vaccinations and seasonality against the progression of the pandemic as measured by deaths or cases. We created a model of all three factors added together to produce a reasonable approximation of the actual progression of the pandemic as measured by deaths or cases. We then removed one of these factors at a time from this model, i.e., NPIs or seasonality to assess how much influence or variation it caused in the original model. If there is a large difference, then this factor could be responsible for a lot of the variation in the progression of the pandemic. If there is little difference, then we infer that this factor has little influence. If the removal of any of these factors approximated a straight line, then this factor was inferred to have a significant bearing on the dynamics of the pandemic.

We carried out a multivariable fitting using all three factors, i.e., NPIs, vaccinations and seasonality (red line) to the progression of the pandemic as measured by time-lagged death (black dotted line) (Figure 9). This provides an estimate of how much of the dynamics and progression of the pandemic for each country was explained in terms of the sum of all three factors. We repeated this multi-regression analysis for cases (Figure 10), positivity rate (Figure 11), hospitalisation (Figure 12) and ICU occupancy (Figure 13). Data for the statistical fittings can be found in the Appendix A.

We find that for all six countries and for all metrics (time-lagged deaths, cases, positivity rates, hospitalisations and ICU occupancy), these combined multivariable fittings are indeed able to capture much of the observed progression of the pandemic in terms of both the timings and magnitudes of the various waves. This can be seen by comparing the observed data (black dotted lines) to the statistical fits of all three factors (red lines) in each panel of Figure 9, Figure 10, Figure 11, Figure 12 and Figure 13. We acknowledge that the fits are not perfect and that there are probably additional factors that influenced the pandemic dynamics and progression over the three-year study period for all six countries. Nonetheless, we think that the fittings using all three factors can collectively explain much of the observed path of the pandemic.

In the previous subsections, we looked at the relative contributions of each of the three factors in isolation by evaluating how much each factor could explain individually. Now, let us assess the relative contributions of each factor in a different manner—by removing each factor in turn from our multivariable regression and evaluating how the fitting changes compared to the combined three-factor regression.

The results from this “withholding” analysis are indicated in each panel in Figure 9, Figure 10, Figure 11 and Figure 12 with different coloured lines. When we look at the effect of the stringency of NPIs against the number of deaths in Figure 3, it is not completely unequivocal (from a visual inspection) if these NPIs have any effect on the reduction or increase in the number of deaths. To a lesser extent, this might also be true of Figure 5, where we plot the stringency of NPIs against the total number of vaccinations in each country. In contrast, we can see a definite pattern when examining the number of deaths against seasonality in Figure 7. In Figure 9, we see that if we remove the influence of seasonality, the variation in the modelled line (black) is very much reduced. In some countries like Sweden, this contrast is quite extreme as it appears to be an almost flat line. In contrast, the statistical fitting that excludes the effects of NPIs (green line) and those that exclude vaccination (blue line) seems to follow the combined modelled line (red) suggesting that they do not contribute much to the variation in the modelled line and therefore they do not explain much of the variation in the progression of the pandemic. There may be some minor differences, e.g., the fits for Finland are slightly different when excluding vaccination rates (blue lines), but they are relatively modest.

This trend varies slightly with each of the metrics we used to gauge the progression of the pandemic. In Figure 10, which uses cases as a metric, there are some instances in which the fitted line that describes the removal of seasonality follows the main statistical fit to all three proposed influences, i.e., vaccinations, NPIs and seasonality. This means that it is difficult to ascribe a lot of influence of seasonality to this specific metric. However, it must be remembered that the measurement of cases was not consistent throughout the entire pandemic given the changes in definitions and testing regimes over time and throughout the six countries. In cases such as Ireland and the UK, the testing regime changed from confirmed RT-PCR tests to a mixture of clinical tests and self-reported antigen tests for the wider community.

This may also be similar to measurements of the pandemic based on the positivity rates. For instance, excluding the contribution of seasonality does not explain some of the variation in the progression of the pandemic for Sweden, Denmark, Finland and Norway (Figure 11), however, the contribution of seasonality seems to be clearer for Ireland and the UK.

The results are broadly similar when we analyse the pandemic progression using either of the other metrics, i.e., hospitalisations (Figure 12) and ICU occupancy (Figure 13). However, the collection of this latter data in some countries was terminated before the end of the pandemic.

Overall, we think that this combination of results confirms the findings of the previous subsections. That is, when seasonality is included as a contributing factor (red, green or blue lines), the multivariable regression can explain much of the observed dynamics of the pandemic over the study period of 2020–2023 (black dotted lines). However, when seasonality is excluded as a contributing factor, the best fits in terms of the other two factors are not especially compelling.

Recent studies by Shamsa et al., 2023, using an entirely different method of study based on spectral analysis and naturally occurring frequencies of oscillation seem to confirm some of our major findings for the USA at least. They stated that part of the reason that “COVID-19 appeared to follow a repeating pattern of outbreaks regardless of social distancing, mask mandates, and vaccination campaign” could be the predictable seasonality of COVID-19 outbreaks [62].

We acknowledge that this analysis only applies to the specific geographical locality of these six countries for this dataset, therefore results should be interpreted with caution when extrapolating to other regions and time periods.

## 4. Study Limitations and Future Research Directions

The NPI stringency index used in this study was developed by Hale et al., 2021 [4] and involves considerable subjectivity. However, this dataset has been used in many peer-reviewed publications and has gone through multiple independent evaluations—see Hale et al., 2021 [4] for details. Although our analysis was confined to six neighbouring countries, some studies have suggested the effects of NPIs can differ between countries [95]. In particular, measuring differences in the degree of compliance to NPIs for each country and over time can be challenging and there is the possibility some differences between the six Northern European countries might potentially have been overlooked in the construction of the series by Hale et al. due to limitations in the publicly available data.

There are also concerns over the heterogenous nature of COVID-19 data collection since the beginning of the pandemic given the changes in definitions, demographics of the population tested and inconsistencies of these measurements between countries over time [22]. We have attempted to overcome some of these concerns by considering a range of metrics including positivity rates, cases, hospitalisation and ICU occupancy when looking for strong signals from potential influences. Other researchers such as Bjørnskov et al. circumvented the heterogenous nature of COVID-19 reporting by using the weekly death statistics rather than deaths with COVID-19 [38]. In contrast, our study explicitly assumes that the recording of “death with COVID-19” in the public dataset is accurate.

To describe coronavirus seasonality, we have used the average of 10 years of empirical data from a recording of the endemic human coronaviruses in Stockholm. However, as can be seen (Figure 1b), the exact seasonality for beta-coronaviruses OC43 and HKU1 varies from year to year. Therefore, future research might attempt to improve a statistical fit by taking interannual variability into account. Future work may also try to complement our statistical empirical analysis with model-based assessments, e.g., estimating the changes in the theoretical instantaneous reproduction number as an additional metric for the progression of the pandemic.

We also note that future studies can potentially be built upon the strengths of our study, which include the use of the complete empirical dataset for COVID-19 and a recent dataset for the incidence of endemic beta-coronavirus infections in Northern Europe. It would also be useful to expand our analysis to other regions and climates or to studies incorporating additional variables that might influence seasonality. A key part of this would be to find an equivalent proxy for seasonality. We have identified the US as one potential region where such an extension might be promising since Kissler et al. [46] have already approximated historical beta-coronavirus incidence in the United States. However, whereas our study area was more geographically confined, the US covers a much larger geographical area with multiple climatic regimes (from subtropical and tropical areas to polar areas), so it might be necessary to first subdivide the US statistics into similar climatic zones.

Finally, this study is limited to six Northern European countries. Although these findings have important implications for future global responses, caution should be taken when extrapolating the analysis to other countries. Additionally, there may be other confounding variables that might be of importance in the dynamics of COVID-19 but which we have not considered in this manuscript.

## 5. Conclusions and Recommendations

The authors sought to examine the progression of the COVID-19 (SARS-CoV-2) pandemic in six Northern European countries (Ireland, the UK, Denmark, Norway, Sweden and Finland) over the duration of the pandemic (March 2020 to May 2023) and assess how much (of the pandemic progression) could be explained in terms of three different factors, namely:The stringency of NPIs, e.g., travel restrictions, “stay-at-home” measures, mask mandates and social distancing;The percentage of the population fully vaccinated with COVID-19 vaccines (defined in terms of the original recommended numbers of doses for each vaccine);The expected seasonality of human beta-coronaviruses (since SARS-CoV-2 is a human beta-coronavirus)—using as a proxy the decadal average of human beta-coronavirus incidences in Sweden based on Swedish clinical data (2010–2020) recorded before the COVID-19 pandemic began.

We created two simple tables to summarise the main results of our analysis of these six countries—Table 1 and Table 2.

Several high-profile studies, some of which had a major influence on health policy throughout the pandemic, have proposed that the various NPIs implemented by governments have been the major (ongoing) factors altering the progression of the pandemic and that the pandemic would have unfolded very differently in the absence of these NPIs [2,3,40,120]. This initially appears to contradict our findings. However, we note that none of these studies finding a strong role for NPIs in the pandemic progression appear to have considered the seasonality of the virus. Moreover, most of these studies confined their analysis to either the first wave [2,3,18,19,20,21] or the second wave [40] of the pandemic.

Indeed, data from the six Northern European countries reveals that changes in the stringency of NPIs generally followed the progression of the pandemic, rather than the other way around. If the NPIs were as effective at altering the progression of the pandemic as claimed, we should expect a uniform negative correlation for the six Northern European countries, but we noted that the correlations were often positive. Furthermore, when the expected negative correlations were identified, they were typically strongest if the interventions lagged the pandemic progression, i.e., the opposite of what was expected.

Meanwhile, several studies have proposed that, if a large fraction of a country’s population received any of the four approved COVID-19 vaccines [5,6,7,8], this should have dramatically altered the progression of the pandemic [11,23,24]. Our findings also appear to contradict this, since, although a widespread population-wide vaccination programme in all six countries succeeded in vaccinating most of the population (all ages) [121], our analysis could not identify any evidence of this hoped-for effect. Again, however, we note that those studies proposing a strong influence of the vaccination programmes on the progression of the pandemic do not appear to have considered the seasonality of the virus.

In contrast to the first two factors, our analysis revealed a striking similarity between the winter highs and summer lows of the COVID-19 pandemic and those of the decadal averages of beta-coronaviruses recorded in Sweden in the preceding decade (2010–2020), even though these were biologically distinct data. This data showed a clear strong positive seasonal correlation, following similar trends for all countries. Therefore, of all these three factors, seasonality appears to have been the dominant driver of the COVID-19 pandemic in these six Northern European countries.

For these reasons, we hypothesise that much of the apparent success of NPIs and vaccination in bringing the pandemic under control might actually have been an effect of seasonality. That is, the increasing stringency of NPIs in spring 2020 and the roll-out of vaccination programmes in spring 2021 may have coincided with the declining incidence of the virus due to the onset of summer.

The authors note that current approaches to the mathematical modelling of epidemics do not typically consider seasonality [63]. This includes modelling studies that have (prematurely) concluded that either NPIs [2,3,21,40] or vaccinations [11] have substantially altered the progression of the COVID-19 pandemic. Therefore, we suggest that it is time for the epidemic modelling community to revisit current modelling techniques to explicitly account for seasonal influence. This should also be considered for future epidemics/pandemics from viruses known to exhibit seasonality, including influenza [47]. Additionally, we highly recommend that policymakers who relied on NPIs and/or vaccination programmes for their COVID-19 response policies revisit their past assumptions about the effectiveness of these policies for the COVID-19 pandemic. We recommend that these findings also be taken into account if similar policies are considered during future large-scale public-health emergencies, especially considering that unintended consequences have been documented for NPIs [66] and vaccinations [121] during the COVID-19 pandemic.

That said, as mentioned in the introduction, we caution that our analysis is a population-based, observational, ecological study that evaluates the progression of the pandemic at a population level only, as distinct from an experimental-based study. Therefore, our results should be interpreted in light of the possibility of the so-called “ecological fallacy”. That is, the results at a population level for each of the six countries might potentially mask significant effects at an individual level. Nonetheless, we believe the results of our current study could still help public-health decision-makers prepare for similar outbreaks of COVID-19 and inform them about the types of policies that were effective in controlling its progression. Great care should also be exercised when extrapolating these results to different geographical areas and socio-economic conditions.

## Figures and Tables

**Figure 1 jcm-13-00334-f001:**
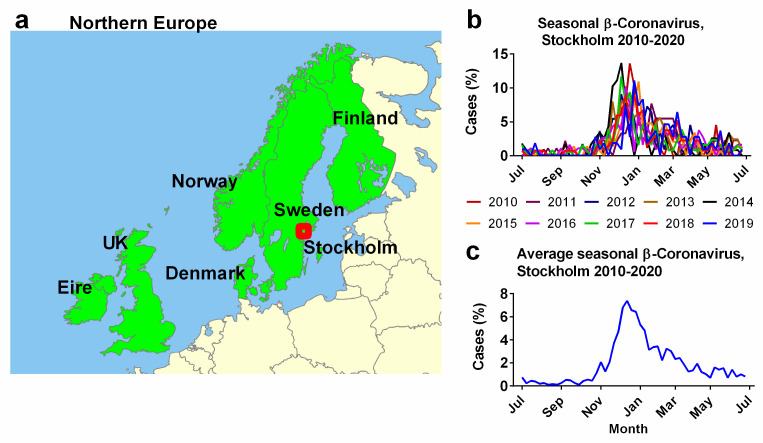
Seasonal influence of human beta-coronavirus in the six Northern European Countries. (**a**) Northern European countries examined in this study as defined by the World Geographical Scheme for Recording Plant Distributions modified from base map created by Rkitko, 2 June 2014. CC BY-SA 3.0. https://commons.wikimedia.org/wiki/File:WGSRPD_Northern_Europe.svg (accessed on 25 July 2023). (**b**) Seasonal incidence of human beta-coronaviruses OC43 and HKU1 from 2010–2020 recorded by the University Hospital, Stockholm, Sweden (red dot, (**a**)). (**c**) Average observations over 10 years (2010–2020) beginning at epidemiological week 26 through to the same week of the next year. Data from Neher et al. (2020) [45].

**Figure 2 jcm-13-00334-f002:**
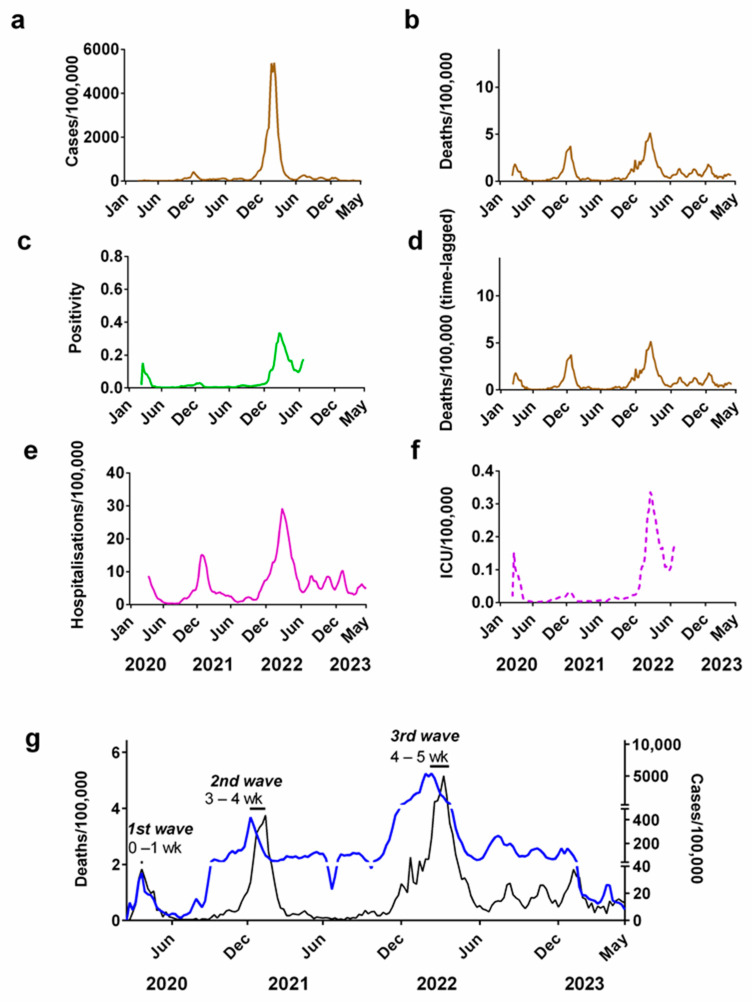
Progression of the COVID-19 pandemic in Denmark. COVID-19 progression measured by: (**a**) cases (per 100,000); (**b**) deaths (per 100,000); (**c**) positivity rate; (**d**) time-lagged death (3-week lag, expressed per 100,000); (**e**) hospitalisations (per 100,000); (**f**) ICU occupancy (per 100,000); (**g**) time lag between cases (blue line) and deaths (black line). Note that the secondary *y*-axis for (**g**) has been split into three scales to allow easier comparison of the peaks of the waves. Data covering the period 1 March 2020 to 6 May 2023 were taken from “Our World in Data” (https://ourworldindata.org/coronavirus; accessed on 25 July 2023 [69]).

**Figure 3 jcm-13-00334-f003:**
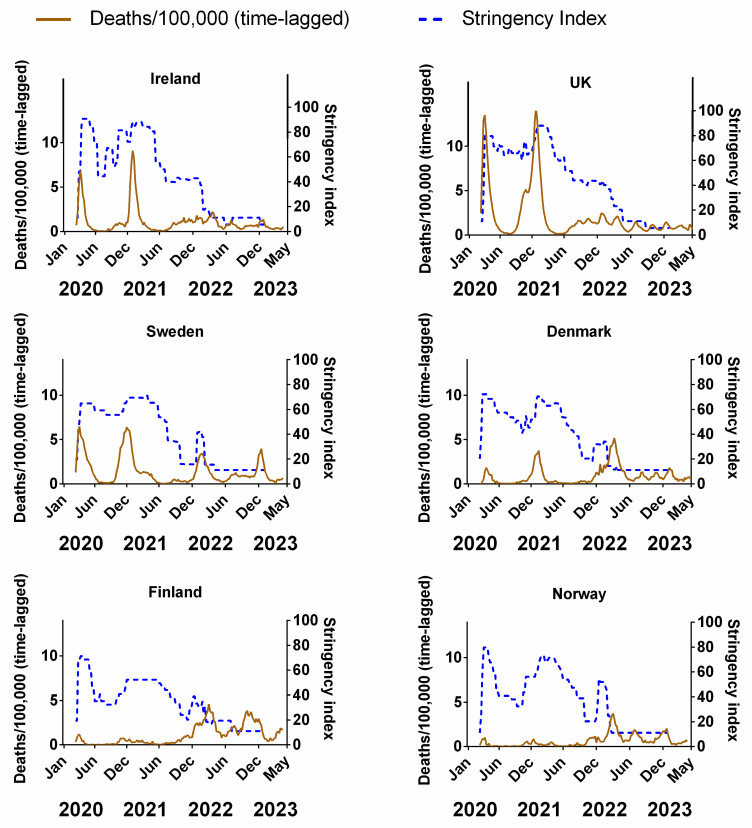
The relative stringency of non-pharmaceutical interventions compared to the progression of the COVID-19 pandemic for each Northern European country. COVID-19 pandemic measured by time-lagged deaths (assuming a 3-week lag between infection and death, expressed per 100,000). Data covering the period 1 March 2020 to 6 May 2023 were taken from “Our World in Data” (https://ourworldindata.org/coronavirus; accessed on 25 July 2023 [69]).

**Figure 4 jcm-13-00334-f004:**
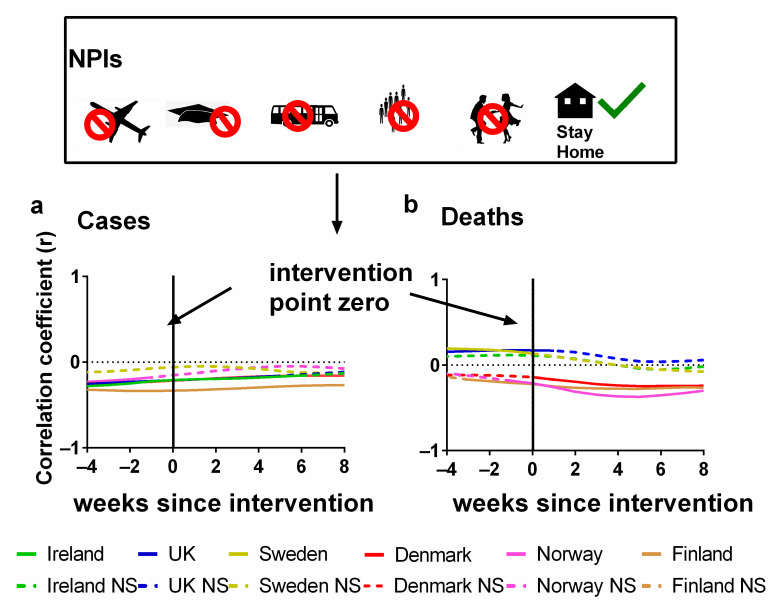
Time-lagged Pearson correlation tests between the stringency of NPIs and the progression of the COVID-19 pandemic. Correlation of the (potential) time-lagged effects of non-pharmaceutical interventions (NPIs) with the progression of the COVID-19 pandemic as measured by (**a**) COVID-19 cases/100,000 or (**b**) COVID-19 deaths/100,000) for weekly time lags. Non-significant values are indicated by NS. Data covering the period 1 March 2020 to 6 May 2023 were taken from “Our World in Data” (https://ourworldindata.org/coronavirus; accessed on 25 July 2023 [69]).

**Figure 5 jcm-13-00334-f005:**
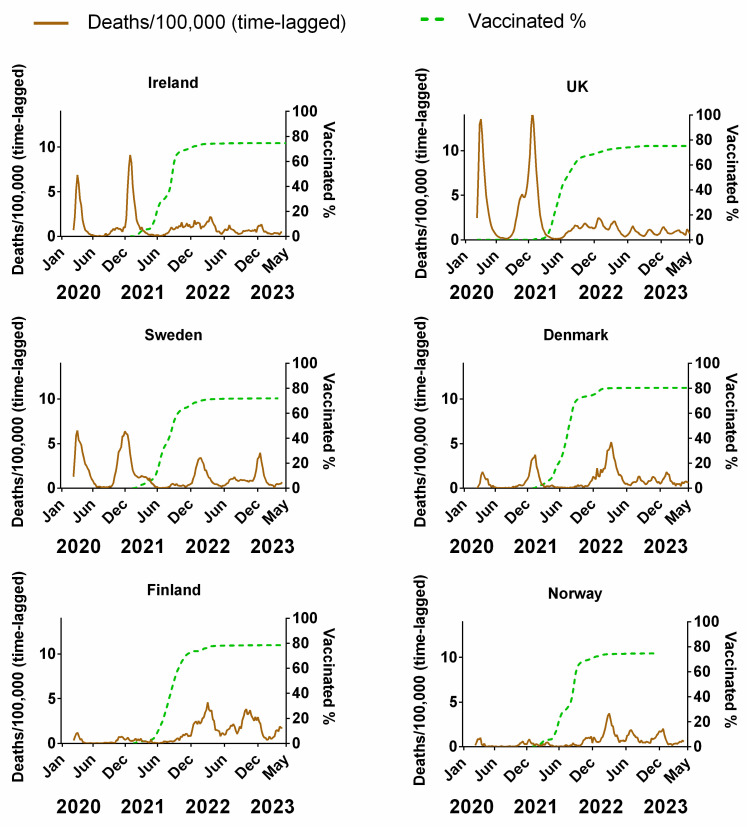
The percentage of the population fully vaccinated compared to the progression of the COVID-19 pandemic for each Northern European country. COVID-19 pandemic progression measured by deaths (assuming a 3-week lag between infection and death, expressed per 100,000). Data covering the period 1 March 2020 to 6 May 2023 taken from “Our World in Data” (https://ourworldindata.org/coronavirus; accessed on 25 July 2023 [69]).

**Figure 6 jcm-13-00334-f006:**
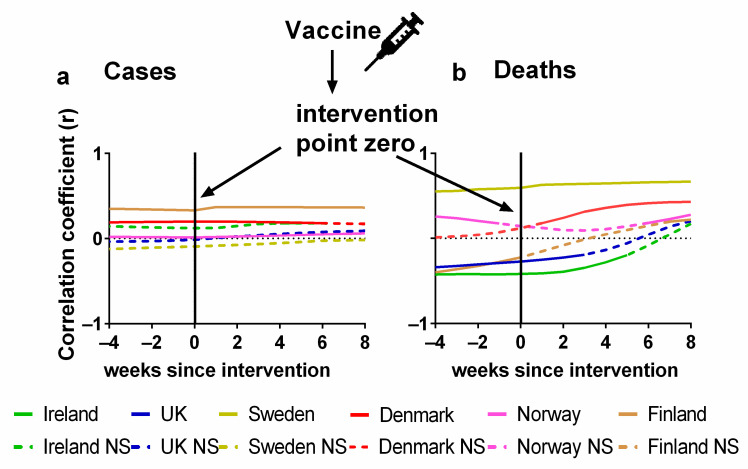
Pearson correlation test of Northern European vaccination time series compared to COVID-19 pandemic progression. Pandemic progression measured by (**a**) COVID-19 cases, (weekly time-lagged cases per 100,000), (**b**) deaths time series, (weekly time-lagged deaths per 100,000). Data covering the period 1 March 2020 to 6 May 2023 were taken from “Our World in Data” https://ourworldindata.org/coronavirus; accessed on 5 July 2023 [69].

**Figure 7 jcm-13-00334-f007:**
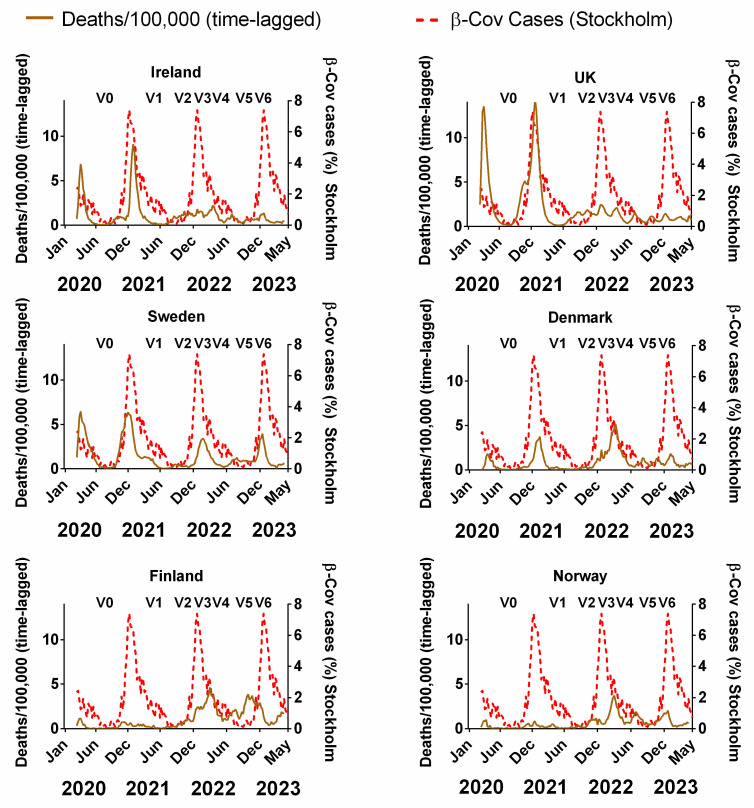
The average seasonal incidence of human beta-coronavirus cases in Stockholm (2010–2020) compared to the progression of the COVID-19 pandemic. COVID-19 pandemic progression measured by time-lagged death (assuming a 3-week lag between infections and deaths, expressed per 100,000). Chronological data on COVID-19 variants, V0 = Wuhan strain, V1 = Alpha, V2 = Delta, V3 = Omicron BA.1, V4 = Omicron BA.2, V5 = Omicron BA.5 and V6 = Omicron BQ.1, sourced from GISAID, via CoVariants.org accessed on 1 November 2023. COVID-19 data covering the period 1 March 2020 to 6 May 2023 were taken from “Our World in Data” (https://ourworldindata.org/coronavirus; accessed 5 July 2023 [69]). The weekly beta-coronavirus (HCoV-OC43 and -HKU1) data (1 January 2010 to 2 April 2020) were recorded by the University Hospital in Stockholm, Sweden [45].

**Figure 8 jcm-13-00334-f008:**
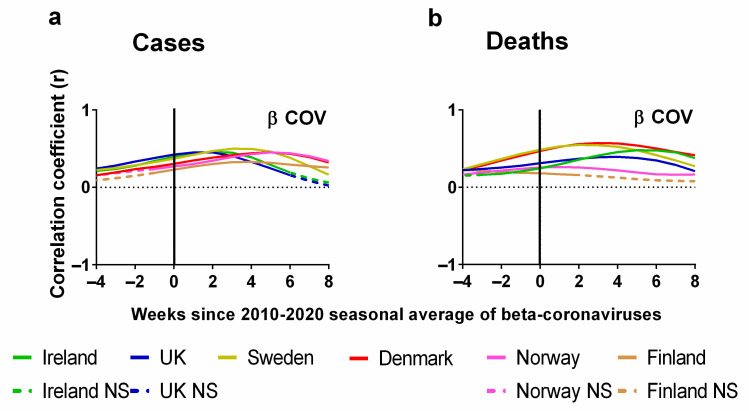
Pearson correlation test of weekly average human beta-coronavirus cases in Stockholm compared to progression of COVID-19 pandemic. Progression of pandemic measured by (**a**) COVID-19 cases (cases per 100,000) and (**b**) deaths (3-week time lag, expressed per 100,000). Data covering the period 1 March 2020 to 6 May 2023 were taken from “Our World in Data” https://ourworldindata.org/coronavirus; accessed on 5 May 2022 [69] and weekly beta-coronavirus (HCoV OC43 and HCoV HKU1) cases (1 January 2010 to 2 April 2020) from the University Hospital in Stockholm, Sweden [45].

**Figure 9 jcm-13-00334-f009:**
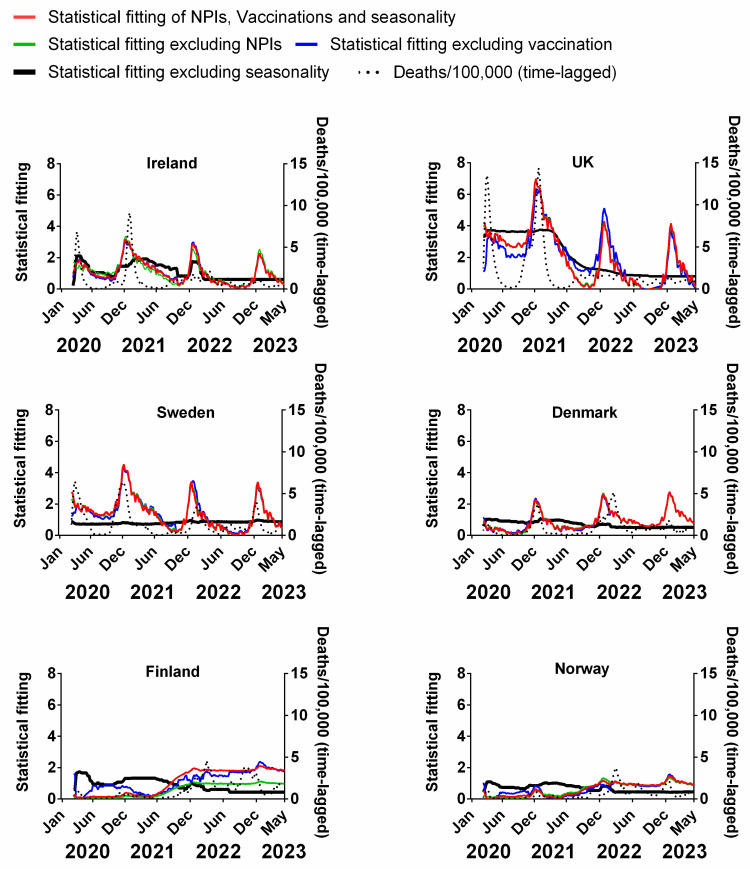
Statistical fitting of NPIs, vaccinations and seasonality to time-lagged deaths in six Northern European countries. Data covering the period 1 March 2020 to 6 May 2023 were taken from “Our World in Data”, https://ourworldindata.org/coronavirus; accessed on 5 July 2023 [69] and weekly beta-coronavirus (HCoV-OC43 and HCoV-HKU1 cases (1 January 2010 to 2 April 2020) from the University Hospital in Stockholm, Sweden [45].

**Figure 10 jcm-13-00334-f010:**
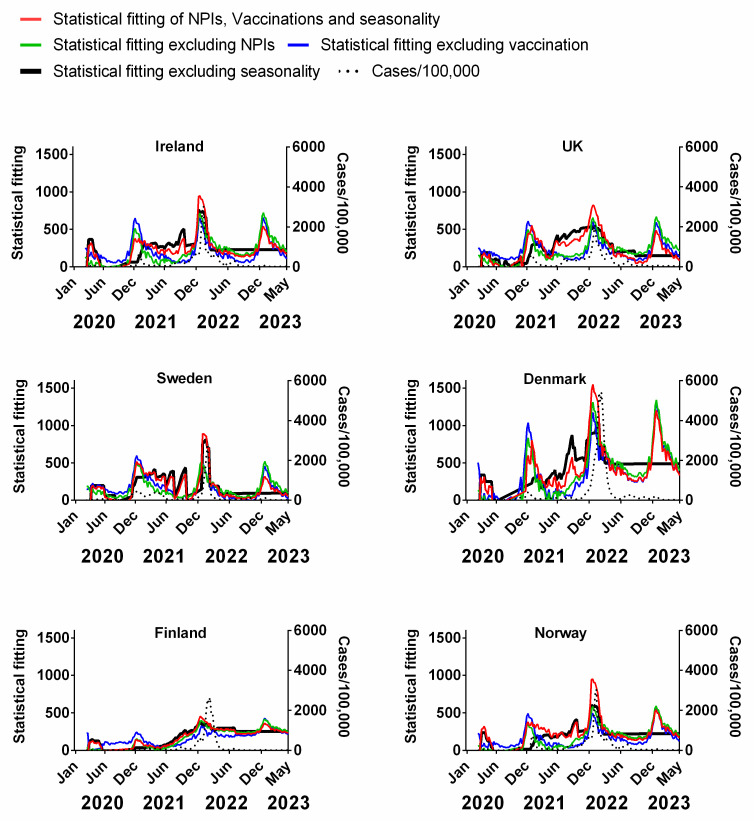
Statistical fitting of NPIs, vaccinations and seasonality to cases of COVID-19 in six Northern European countries. Data covering the period 1 March 2020 to 6 May 2023 were taken from “Our World in Data”, https://ourworldindata.org/coronavirus; accessed on 5 July 2023 [69] and weekly beta-coronavirus (HCoV-OC43 and HCoV-HKU1) cases (1 January 2010 to 2 April 2020) from the University Hospital in Stockholm, Sweden [45].

**Figure 11 jcm-13-00334-f011:**
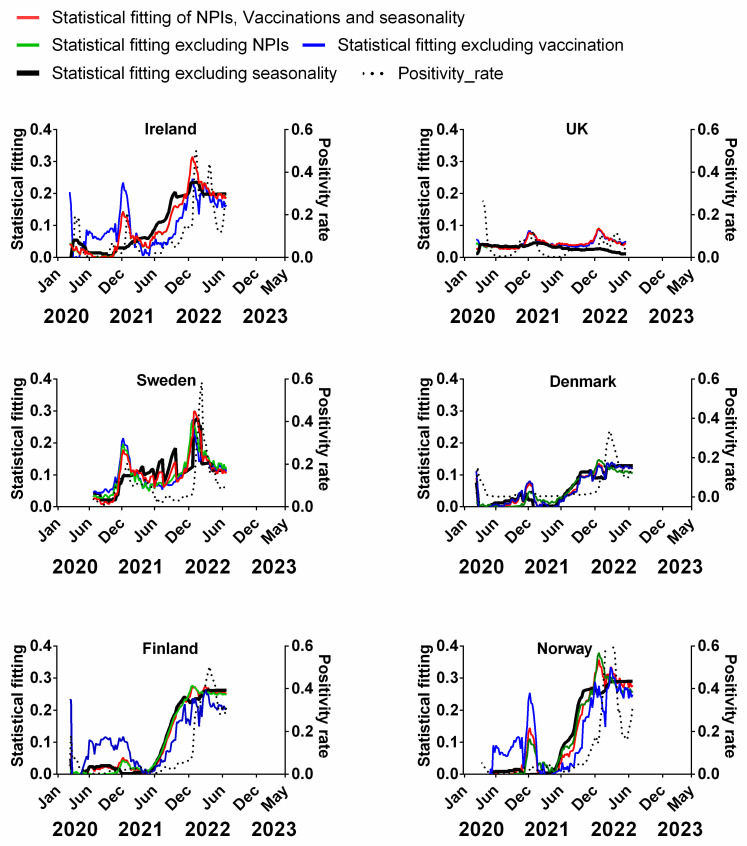
Statistical fitting of NPIs, vaccinations and seasonality to positivity rate in six Northern European countries. Data covering the period 1 March 2020 to 6 May 2023 were taken from “Our World in Data”, https://ourworldindata.org/coronavirus; accessed on 5 July 2023 [69] and weekly beta-coronavirus (HCoV-OC43 and HCoV-HKU1) cases (1 January 2010 to 2 April 2020) from the University Hospital in Stockholm, Sweden [45].

**Figure 12 jcm-13-00334-f012:**
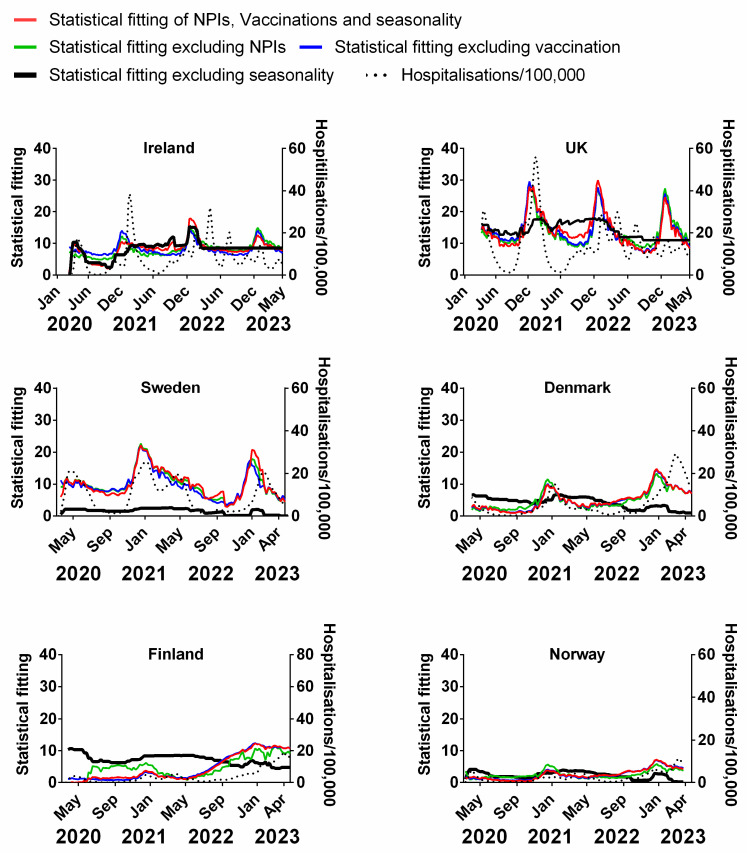
Statistical fitting of NPIs, vaccinations and seasonality to hospitalisations in six Northern European countries. Data covering the period 1 March 2020 to 6 May 2023 were taken from “Our World in Data”, https://ourworldindata.org/coronavirus; accessed on 5 July 2023 [69] and weekly beta-coronavirus (HCoV-OC43 and HCoV-HKU1) cases (1 January 2010 to 2 April 2020) from the University Hospital in Stockholm, Sweden [45].

**Figure 13 jcm-13-00334-f013:**
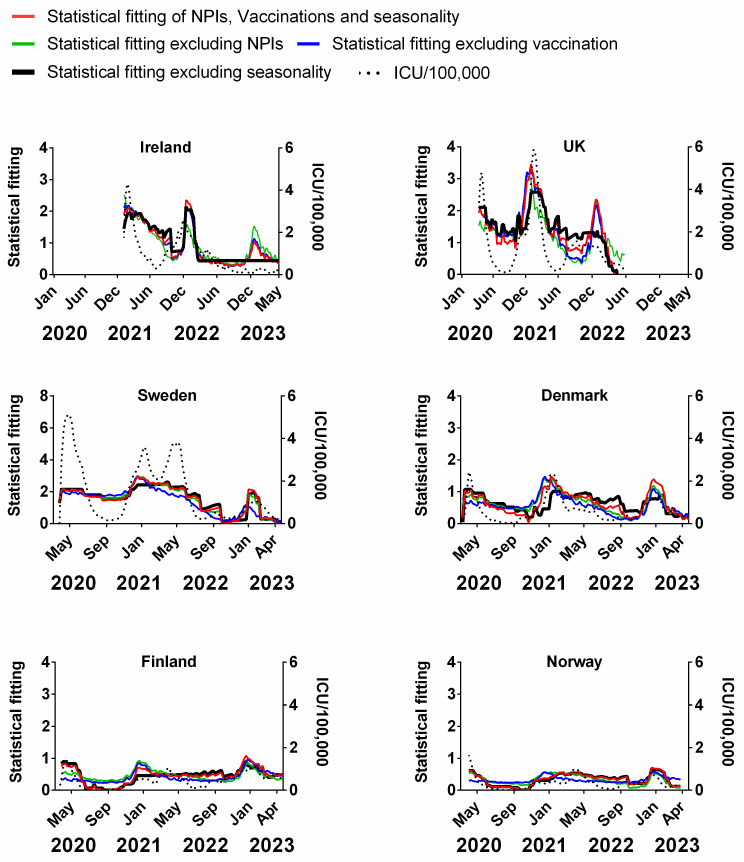
Statistical fitting of NPIs, vaccinations and seasonality to ICU occupancy in six Northern European countries. Data covering the period 1 March 2020 to 6 May 2023 were taken from “Our World in Data”, https://ourworldindata.org/coronavirus; accessed on 5 July 2023 [69] and weekly beta-coronavirus (HCoV-OC43 and HCoV-HKU1) cases (1 January 2010 to 2 April 2020) from the University Hospital in Stockholm, Sweden [45].

**Table 1 jcm-13-00334-t001:** Influence on progression of pandemic in terms of lagged deaths.

Northern European Countries	NPIs	Vaccination	Seasonality
Ireland	**x**	**x**	**✓**
UK	**x**	**x**	**✓**
Sweden	**x**	**x**	**✓**
Denmark	**x**	**x**	**✓**
Finland	**x**	**x**	**✓**
Norway	**x**	**x**	**✓**

✓ in green background indicates that a clear, consistent and physically plausible influence of the factor on the progression of the pandemic was identified for this country while x in red background indicates that an influence was not identified for this country.

**Table 2 jcm-13-00334-t002:** Influence on progression of pandemic in terms of cases.

Northern European Countries	NPIs	Vaccination	Seasonality
Ireland	**x**	**x**	**✓**
UK	**x**	**x**	**✓**
Sweden	**x**	**x**	**✓**
Denmark	**x**	**x**	**✓**
Finland	**x**	**x**	**✓**
Norway	**x**	**x**	**✓**

✓ in green background indicates that a clear, consistent and physically plausible influence of the factor on the progression of the pandemic was identified for this country while x in red background indicates that an influence was not identified for this country.

## Data Availability

The source data for this manuscript are publicly available from Our World in Data”, https://ourworldindata.org/coronavirus; accessed on 5 July 2023. The data on beta-coronavirus (HCoV-OC43 and HCoV-HKU1) cases (1 January 2010 to 2 April 2020) are described in the reference [45] and were kindly provided to us by Dyrdak and Albert. Other data generated from these are available in Supplementary Excel File S1.

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
