# Peer review of "Influence of Seasonality and Public-Health Interventions on the COVID-19 Pandemic in Northern Europe"

_jcm, 2024, doi:10.3390/jcm13020334_

Round 1

Reviewer 1 Report

Comments and Suggestions for Authors

This is an extremely interesting manuscript and it deals with a critical issue of response to the global COVID19 pandemic, with the limitation that is focused on a Upper middle income/high income subset of countries. This has important implications for future global responses.

I appreciate the thoroughness and creative analytical approaches used by the authors, and the willingness to challenge paradigms of “what works.” I appreciate that the variables used were well described and documented (e.g. using an established NPI index, sourcing seasonality data, etc.)  

I do however have several concerns:

1.      I am concerned about the structure of the manuscript. In reviewing the instructions to authors, the format should be “Introduction, Materials and Methods, Results, Discussion, Conclusions (optional).” However, your paper combines the results and discussion in the section marked “results” (to the point that I had to check if there was an alternative format used by this journal). This likely made the presentation less effective as the authors enter into a discourse after the presentation of each variable arguing the relative merits and drawbacks. While I appreciated the feeling this contributed of being in a tabletop discussion, it also made the manuscript very long. The convention of separating results and discussion serves to make us present a concerted, focused argument after reviewing all the evidence, which would be useful here.   

2.      There is a repetition. In one example, there is convincing presentation of the weakness of case counts as a useful outcome, and discussion that these will be left as supplemental materials, yet case counts are used in several of the primary presented analyses.  As well the principle of lack of evidence is not equal to evidence of nonexistence (a point well made) is repeated several times.

3.      NPIs -- The authors make a valid point that NPIs are often implemented as a political response vs in a truly “preventive” manner. This plotting of deaths/epidemic activity alone is a finding that is useful. This does, then call into question whether the variable as such is able to function as intended in the multivariate analysis, if you have already documented that the univariate behavior was consistent. The effectiveness of implementation of the NPIs is not necessarily measured – again, this may be a difference from LMICs where I spent the pandemic and Northen Europe, but one wonders if the compliance with NPIs over time within the country is impactful to the progression of the pandemic over time.

4.      Vaccines – not discussed and potentially impactful is the way in which SCV2 vaccines became available and were rolled out. Often they were only available in limited quantities, especially early in the response, as well acceptability and uptake of the vaccines were issues in some countries (I don’t know how much an issue this may have been in northern Europe, it was significant in LMICs). While the use of % coverage should be sufficient to account for the role of these factors, it is worth considering whether a cumulative measure of vaccine coverage is informative as the pandemic continued. The authors note that booster doses were not considered, however if the population coverage was primarily derived in the first month of vaccine availability this may have been partly explanatory of the higher case count and even deaths with later waves.  

It is also worthwhile to consider whether some of the aspects of the vaccine analysis have been overtaken by events. While there was substantial hope that the vaccine would reduce symptomatic illness when initially introduced, the subsequent experience of breakthrough infections changed the understanding of the role of the vaccine. The authors could be best served by presenting the present understanding of SCV2 vaccines in examining the correlation data with vaccines.

5.      Seasonality – The argument is well made that “waves” of virus transmission are a critical factor in understanding the progression of the pandemic, and I think you have generally argued that this is in line with the other coronaviruse seasonality. It would have been useful to see a comparison of the introduction of SCV2 variants to the region alongside the seasonality data used. The authors at one point refer to the seasonality as a “natural” phenomenon however the measurement is of course human cases, and these are affected by human behavior in response to the season. Does the seasonability of influenza or other respiratory viruses work as well, or there is a specific aspect to the beta coronaviruses?  

6.      I am not convinced that the analytical methods employed are ideal for the aims of this analysis. The temporal changes in management of patients and variant strains make it difficult to use a country’s deaths in 2020 as a fair comparison to evaluate for their deaths in 2022. It might have been interesting (though, I realize, an entirely different paper) to look at inter-country comparisons with these same principle analyses. As the authors repeatedly mention, the negative lags are in many cases unrealistic. While the authors draw the conclusion that the NPIs and vaccines were likely shown to be ineffective, one will also ask whether than analytical method is the most apt. I have suggested that the editors seek a detailed statistical reviewer to assist with this point.

Reviewer 2 Report

Comments and Suggestions for Authors

The article "The influence of non-pharmaceutical interventions, vaccination, and coronavirus seasonality on the progression of the COVID-19 pandemic in Northern Europe" proposed by Quinn G. et al. addresses a highly relevant topic. After a careful reading, I have identified the sections that require improvement, which I detail below.

Summary: The content of the document can be optimized by placing more focus on several key aspects. It would be beneficial to start by providing a clearer context, followed by outlining the main objectives. Subsequently, it is crucial to detail the methodology used, specifying the type of study conducted and providing information on the number of participants involved. Finally, it is recommended to highlight in a timely manner the main results obtained, and the conclusions derived from these findings. This approach will contribute to a more structured and comprehensible presentation of the work.

It is necessary to perform a thorough review of the keywords, paying special attention to their concordance with MeSH (Medical Subject Headings) terms. This process will ensure greater accuracy and alignment with the standard terminology used in the scientific literature, thus improving the quality and relevance of the selected keywords.

Introduction

It is imperative to provide a more detailed elaboration of the information presented in the study, focusing specifically on the situation in Northern Europe. It is suggested to concisely summarize previous studies related to the progression of the pandemic. In addition, it is essential to highlight gaps in current understanding and explain how this study intends to address those gaps.

Regarding acronyms used throughout the article, it is recommended that a thorough review be conducted and, as a first instance, provide the full definition of each acronym (e.g., starting with the World Health Organization, WHO). This will facilitate the reader's understanding and improve the clarity of the document.

Methodology

I recommend that the glossary section be presented as supplementary material. In the organization of the study, it is crucial to specify the design used and provide details on the overall sample size. The omission of repetitions of information should be carefully checked, especially in relation to data extraction, as this aspect has already been addressed in the introduction.

Results and Discussion:

The sections of the paper have been structured in an organized manner to facilitate understanding of the information. It is essential to support the findings presented by including more recent studies. Ultimately, it is suggested that a hypothesis be formulated to explain the variations observed in the results compared to previously published studies.

I would recommend that the study incorporate a summary table that concisely highlights key findings related to nonpharmaceutical interventions, vaccination, and coronavirus seasonality in the evolution of the COVID-19 pandemic in northern Europe. This addition would not only simplify the presentation of data but also significantly increase the visual impact and accessibility of the information for the reader.

In the limitations section, it would be beneficial to highlight more emphatically the inherent strengths of the study. Also, future perspectives could be addressed that, although not addressed in the present analysis, could be valuable focal points for further research.

Figures: Good quality

Bibliographic references are updated.

Reviewer 3 Report

Comments and Suggestions for Authors

Dear authors,

Your manuscript titled “The Influence of Non-pharmaceutical Interventions, vaccination, and Coronavirus Seasonality on the Progression of the COVID-19 Pandemic in Northern Europe” and its associated abstract and keywords offer a comprehensive overview of your study. Here are some suggestions for improvement:

Title

  1. Clarity and Conciseness: The title is informative but slightly lengthy. Consider shortening it while maintaining its essence. For example: “Impact of Non-Pharmaceutical Interventions, Vaccination, and Seasonality on COVID-19 in Northern Europe.”
  2. Specificity: The title broadly covers three significant factors. If one of these factors is the primary focus or shows the most critical findings, consider highlighting it more prominently.

Abstract

  1. Background: The background information is clear and sets a suitable context. However, adding a sentence about why these three factors (NPIs, vaccination, and seasonality) were chosen for the study might provide more insight into the rationale behind the study.
  2. Methods: The methods are well-described, but consider briefly mentioning the statistical techniques used for analysis (e.g., Pearson correlation coefficients) here for completeness.
  3. Findings: The findings are intriguing, especially the unexpected results regarding the correlations. Ensure that the language is cautious and reflects the study’s observational nature. It might be helpful to specify that these are correlations, not causations.
  4. Implications: The implications are well stated but could be strengthened by explicitly stating the potential impact on future policy decisions.
  5. Limitations: Consider adding a sentence about the limitations of your study. This adds to the credibility and helps readers understand the context of your findings better.

Keywords

  1. Relevance: The keywords are relevant. However, consider adding a few more specific to the regions studied (e.g., “Northern Europe,” “Ireland,” “UK,” etc.) to enhance discoverability.
  2. Specificity: Adding keywords related to the specific methodologies or statistical analyses might attract a more targeted audience.

The abstract summarizes the key points well, but ensuring clarity, especially in conveying the nature of your findings (correlational, not causal), is crucial. The title, while informative, could be made more concise for better impact.

Introduction

Your introduction to the manuscript provides a detailed and well-structured overview of the context and rationale for your study. Here are some suggestions for improvement:

Structure and Flow

  1. Contextual Setting: The introduction starts with a historical perspective, which is effective. However, it could benefit from a brief mention of the global impact of COVID-19 to set a broader context.
  2. Transitions Between Points: Ensure smooth transitions between different sections. For instance, the shift from discussing vaccination efficacy to the seasonal nature of coronaviruses could be more fluid.

Content and Clarity

  1. Detailing the Rationale: While the introduction does an excellent job of outlining the three main factors studied (NPIs, vaccination programs, and seasonality), the reasoning behind choosing these specific factors could be more explicit. Explaining why these were prioritized over other potential factors would strengthen the rationale.
  2. Addressing Counterarguments: There is an excellent discussion of various viewpoints on the pandemic’s dynamics. It might be beneficial to briefly mention how your study addresses or challenges these existing perspectives.
  3. Clarity on Assumptions: When discussing the assumptions made in previous studies, clearly outline how your approach differs or improves upon these methodologies.

Technical Details

  1. Data Sources and Justification: The mention of data sources is good, but a brief justification for choosing these sources (e.g., “Our World in Data,” Neher et al. dataset) would add credibility.
  2. Statistical Methods: There is a mention of multivariate statistical analysis, but more detail on the type of statistical methods or models used might be helpful for the reader.

Relevance and Implications

  1. Broader Implications: The introduction does well in setting up the study’s importance. Strengthen this by briefly mentioning the potential broader implications of your findings.
  2. Limitations and Geographical Focus: While the geographical focus is justified, acknowledging the limitations of this focus (e.g., how findings might differ in other regions) would be beneficial.

Language and Tone

  1. Objective Tone: Ensure that the language remains objective and neutral, especially when discussing controversial or debated topics in management.
  2. Consistency in Terminology: Maintain consistency in terms (e.g., COVID-19, SARS-CoV-2) throughout the introduction.

References and Citations

  1. Timely and Relevant Citations: Your introduction includes numerous citations, which is excellent. Ensure that these references are up-to-date and relevant to the points being made.

Your introduction is thorough and sets a solid foundation for the study. Enhancing clarity, especially in the rationale and methodology, and ensuring a smooth flow of ideas will further strengthen this section.

Materials and method

Your manuscript’s’’ “Materials and Methods” section comprehensively explains the methodologies, data sources, and analytical techniques used in your study. Here are some suggestions to enhance this section:

Clarity and Detail

  1. Glossary: Including a glossary is helpful, but consider providing more detail for each term, especially if there is variation in definitions across the countries studied.
  2. Data Sources Justification: While you have mentioned the use of data from “Our World in Data” and Neher et al. (2020), a brief justification for choosing these sources (e.g., reliability, comprehensiveness) would add to the robustness of your methodology.

Consistency and Standardization

  1. Standardizing Metrics: You have outlined the variability in metrics like confirmed cases, deaths, and positivity rates across different countries. Addressing how this variability was managed or standardized in your analysis is essential to ensure comparability.
  2. Inconsistencies in Data: You have acknowledged inconsistencies in data collection and reporting. Clarify how these inconsistencies were addressed or factored into the analysis to maintain the integrity of your findings.

Statistical Analysis

  1. Detailed Explanation of Statistical Methods: While you have mentioned using Pearson correlation coefficients and multiple regression analysis, providing more detail on the rationale behind choosing these methods and how they were explicitly applied would be beneficial.
  2. Addressing Potential Biases: Discuss any potential biases or limitations in your statistical approach and how they were mitigated.

Study Area

  1. Justification of Geographical Focus: You have justified the focus on six Northern European countries. Further elaborating on why these countries provide a representative or particularly insightful sample for this study would strengthen this choice.

Visualization and Supplementary Materials

  1. Accessibility and Usefulness of Supplementary Materials: Make sure that the supplementary materials are not just additional but also meaningful and provide a clear added value to the study’s main findings.

Ethical Considerations

  1. Data Privacy and Ethics: If there are any ethical considerations, especially regarding data privacy or patient information, ensure these are addressed, and compliance with relevant guidelines is clearly stated.

Language and Accessibility

  1. Technical Language: While technical language is necessary, ensure it remains accessible to readers who may not be specialists in every aspect of your study. Avoid overly complex jargon where possible.

Your section is well-structured and thorough. Enhancing clarity, particularly in how data inconsistencies are addressed and why specific statistical methods were chosen, will further improve the robustness and readability of this section.

Results

3.1. Influence of non-pharmaceutical interventions on pandemic progression in Northern Europe

Clarity and Interpretation of Results

  1. Simplification of Language: While the analysis is comprehensive, simplifying the language and making the findings more accessible to a broader readership would enhance the section’s impact. Avoiding overly technical terms where possible will help.

Analysis and Discussion of Results

  1. Contextualizing Negative Correlations: Discussing why negative correlations for certain countries might not imply causation is essential. Expanding on this and providing possible explanations for these observations would strengthen the analysis.
  2. Cautious Interpretation of Correlations: It is good to be careful about interpreting correlations as causations. Emphasizing this point helps maintain scientific rigor.

Comparative Analysis

  1. Comparisons with Other Studies: You have acknowledged that your findings differ from some studies and align with others. Providing a more detailed comparison with these studies, including why there might be discrepancies, would add depth to your analysis.
  2. Limitations of the Study: It is good that you mention the potential limitations of your analysis method. Expanding on these limitations, including how they might affect your findings, is crucial for a balanced view.

Methodological Consistency

  1. Consistency in Metrics Used: Ensure consistency in the metrics used for comparison across different countries and periods. This helps in maintaining the integrity and comparability of the analysis.

Ethical and Responsible Reporting

  1. Avoiding Overgeneralization: Be careful not to overgeneralize the findings. Acknowledging that the results may not universally apply to all settings or populations is essential.
  2. Addressing Potential Biases: Discuss any potential biases in your analysis and how they were mitigated. This could include biases in data sources, statistical methods, or interpretation of results.

Implications and Future Research

  1. Implications of Findings: Discuss the broader implications of your findings for public health policy and future pandemic responses.
  2. Recommendations for Future Research: Suggest areas for further research, particularly in exploring the complex interplay of factors influencing pandemic dynamics.

3.2. Influence of vaccinations on pandemic progression in Northern Europe

Your manuscript’s subsection “3.2. Influence of vaccinations on pandemic progression in Northern Europe” critically analyzes the relationship between vaccination rates and pandemic progression. Here are some suggestions for improvement:

Clarity and Presentation of Data

  1. Simplifying Language: The narrative is quite dense and technical. Simplifying the language and breaking down complex sentences would improve readability and accessibility.

Analysis and Interpretation

  1. Balanced Interpretation of Correlations: Maintaining a balanced view is essential while discussing the correlation (or lack thereof) between vaccination rates and pandemic progression. Acknowledge the complexities and potential confounding factors that might influence these correlations.
  2. Discussion of Alternative Explanations: Expand on alternative explanations for the observed trends. For example, discuss how behavioral changes, natural immunity, or variant evolution might have influenced the outcomes.

Comparative Analysis

  1. Comparative Context: Place your findings with other studies, emphasizing how your methodology, data, or interpretation differs. This can help readers understand why there might be differences in conclusions.
  2. Critical Review of Other Studies: When discussing studies with contrasting findings, critically review their methodologies and assumptions. This will strengthen the rationale behind your conclusions.

Methodological Considerations

  1. Addressing Limitations: Clearly outline any limitations in your analysis, such as potential biases in data sources, the representativeness of the sample, or limitations of the statistical methods used.
  2. Potential Biases: Discuss potential biases in interpreting data, such as confirmation bias, and how they were mitigated in your study.

Implications and Future Research

  1. Broader Implications: Discuss the wider implications of your findings for public health policy, especially regarding vaccination strategies.
  2. Recommendations for Future Research: Suggest areas where further research is needed. This might include studies with different methodologies, longer-term analyses, or research in different geographical settings.

Ethical and Responsible Reporting

  1. Avoiding Overgeneralization: Be cautious not to overgeneralize the findings. Acknowledge that results may not be universally applicable to all settings or populations.
  2. Ecological Fallacy: You have mentioned the ecological fallacy – further emphasize why population-level findings may not necessarily apply to individual cases.

While your analysis is comprehensive, enhancing clarity, providing a balanced interpretation, and situating your findings within the broader research context will strengthen this section of your manuscript.

3.3. Influence of seasonality on pandemic progression in Northern Europe

In the subsection “3.3. Influence of seasonality on pandemic progression in Northern Europe,” you have provided a detailed analysis of the role of seasonality in the progression of the COVID-19 pandemic. To enhance this section, consider including the following elements:

Expanded Analysis and Context

  1. Broader Literature Review: Incorporate a more extensive review of existing literature on the seasonality of respiratory viruses, especially studies that specifically focus on SARS-CoV-2.
  2. Comparative Analysis: Compare the seasonality of COVID-19 with other respiratory viruses like influenza and common cold coronaviruses to provide context and deeper understanding.

Methodological Details

  1. Justification of Methodology: Explain why Stockholm’s decadal average of beta-coronaviruses was chosen as a proxy for seasonality. Discuss its representativeness for Northern Europe.
  2. Statistical Analysis: Provide more details on the statistical methods used to establish the correlation between COVID-19 progression and seasonality. Discuss any limitations or assumptions inherent in this approach.

Interpretation and Implications

  1. Cautious Interpretation: Emphasize the cautious interpretation of correlations and the potential for confounding factors. Discuss how other variables not accounted for in the study might influence the observed patterns.
  2. Broader Implications: Discuss the implications of these findings for public health policy, especially for future pandemic preparedness and response strategies.

Additional Data and Correlations

  1. Environmental Factors: Discuss how temperature, humidity, and UV radiation may influence virus transmission and survival, contributing to seasonality.
  2. Behavioral Factors: Discuss how human behavior changes with seasons (e.g., indoor crowding in winter) may impact the transmission of respiratory viruses.

Comparative Geographical Analysis

  1. Variability Across Regions: Explore the potential for variability in seasonality patterns between the different Northern European countries, considering geographical and climatic differences.
  2. Global Context: Briefly mention how seasonality patterns observed in Northern Europe might differ from other regions, adding a global perspective.

Future Research Directions

  1. Recommendations for Future Research: Suggest areas for future research, such as longitudinal studies across multiple regions and climates or studies incorporating additional variables that might influence seasonality.

Ethical and Responsible Reporting

  1. Avoid Overgeneralization: Ensure that conclusions are specific to the data and contexts analyzed and avoid overgeneralizing the findings to other regions or contexts.
  2. Acknowledging Uncertainties: Acknowledge the uncertainties and complexities in understanding the seasonality of COVID-19 and the need for ongoing research.

While the current analysis provides valuable insights, expanding the context, deepening the methodological explanation, and cautiously interpreting the results will further strengthen this section of your manuscript.

3.4. Influence of multiple factors on pandemic progression in Northern Europe

The subsection “3.4. Influence of multiple factors on pandemic progression in Northern Europe” in your manuscript provides a comprehensive analysis using multivariate regression to understand the combined effects of NPIs, vaccinations, and seasonality on the COVID-19 pandemic. Here are some areas that could be improved or expanded upon:

Clarity and Simplification

  1. Simplify Statistical Explanation: The explanation of the multivariate regression analysis is quite technical. Simplifying the language and breaking down complex statistical concepts would make the findings more accessible to a broader audience.

Methodological Detail

  1. Justification of Methodology: Explain why multivariate regression was chosen as the appropriate method and discuss its advantages and limitations in this context.
  2. Handling of Confounding Factors: Address how the analysis accounts for potential confounding factors and interactions between the three variables (NPIs, vaccinations, and seasonality).

Comparative Analysis

  1. Comparison with Single Factor Analysis: Provide a more precise comparison between the results of the multivariate analysis and the single factor analyses from previous sections. This would help to highlight the added value of considering multiple factors together.
  2. Consideration of Other Factors: Discuss other potential factors that might influence pandemic progression, which were not included in the analysis.

Interpretation and Implications

  1. Cautious Interpretation of Results: Emphasize the cautious interpretation of the statistical findings, especially regarding the causality of the relationships.
  2. Broader Implications: Discuss the broader implications of these findings for public health policy, especially in designing intervention strategies that consider multiple factors.

Data Consistency and Quality

  1. Consistency in Metrics Used: Ensure consistency in the metrics used for comparison across different countries and periods.
  2. Addressing Data Limitations: Acknowledge and discuss any limitations in the data used, such as changes in testing regimes and reporting standards over time.

Future Research Directions

  1. Recommendations for Future Research: Suggest areas for future research, such as exploring additional factors, using different statistical methods, or conducting longitudinal studies.

Ethical and Responsible Reporting

  1. Avoid Overgeneralization: Avoid overgeneralizing the findings and ensure that conclusions are specific to the data and contexts analyzed.
  2. Acknowledging Uncertainties: Highlight the uncertainties and complexities in understanding the multifaceted nature of pandemic progression.

Enhancing clarity, providing a balanced interpretation, and situating your findings within the broader research context will further strengthen this section of your manuscript.

Study limitations

Your manuscript’s “Study Limitations” section critically reflects your study’s constraints and potential biases. Here are some recommendations for further improvement:

Geographic Scope

  1. Broader Geographic Representation: Acknowledge that the findings are based on a specific set of Northern European countries and may not be generalizable to regions with different socio-economic, cultural, or climatic conditions.
  2. Comparison with Other Regions: Suggest how future studies could incorporate data from a more diverse set of countries or regions to validate or contrast the findings.

Data Collection and Consistency

  1. Data Heterogeneity: More explicitly address the challenges and limitations posed by the heterogeneous nature of COVID-19 data collection, such as differences in testing capacities, strategies, and reporting standards.
  2. Alternative Data Sources: Discuss the potential of utilizing alternative or additional data sources to overcome these limitations.

Methodological Considerations

  1. NPI Stringency Index: While acknowledging the subjectivity in the NPI stringency index, discuss the implications of this subjectivity on your findings and how future research might approach this challenge.
  2. Seasonality Data: Reflect on the limitations of using decade-old data to represent current virus seasonality and suggest ways for future studies to update or refine this approach.

Statistical Analysis

  1. Statistical Limitations: Discuss any limitations of the statistical methods, such as the potential for overfitting in multivariate regression or the assumptions underlying the Pearson correlation.
  2. Interannual Variability: Emphasize the importance of considering interannual variability in future research, especially climate change and its potential impact on infectious disease patterns.

Future Research Directions

  1. Complementary Models: Suggest integrating empirical data with theoretical models, such as simulations of virus transmission under different intervention strategies, to provide a more comprehensive understanding.
  2. Additional Factors: Recommend exploring other factors not included in the study but could influence pandemic dynamics, such as public compliance with NPIs, socio-economic factors, or healthcare capacity.

Ethical and Responsible Reporting

  1. Cautious Interpretation: Reiterate the importance of carefully interpreting the results due to the mentioned limitations and avoid overstating the findings.
  2. Transparency: Ensure transparency in reporting the limitations to reinforce the credibility of the study and its findings.

Incorporating these elements will enhance the completeness and credibility of the “Study Limitations” section, providing readers with a clear understanding of the scope and boundaries of your study’s conclusions.

Conclusion and recommendations

Your manuscript’s “Conclusions and Recommendations” section summarizes your findings and their implications. Here are some suggestions for improvement:

Clarity and Conciseness

  1. Simplify Language: Aim for clarity and simplicity in language to make the conclusions accessible to a broader audience, including those outside the specific field of epidemiology.
  2. Summarize Key Findings: Summarize the key findings before delving into their implications. This helps in reinforcing the main points for the reader.

Balanced Interpretation

  1. Cautious Interpretation of Results: While discussing the apparent lack of impact of NPIs and vaccinations based on your study, emphasize that this is specific to the data and context analyzed. Highlight the complexities and potential confounding factors in interpreting these results.
  2. Acknowledging Contrasting Evidence: Recognize the existence of studies with contrasting findings and suggest that your results add to a diverse body of evidence requiring further exploration.

Implications and Recommendations

  1. Policy Implications: Discuss the implications of your findings for public health policy, especially in designing and implementing NPIs and vaccination strategies.
  2. Modeling Techniques: Recommend specific ways to improve epidemic modeling, such as incorporating seasonal variations or other factors overlooked in previous models.
  3. Consideration of Unintended Consequences: Briefly mention the potential unintended consequences of NPIs and vaccination policies, such as economic impacts or social disruption.

Future Research Directions

  1. Suggest Further Research: Recommend areas for future research, such as longitudinal studies, studies in different geographic contexts, or research incorporating additional variables that might influence pandemic dynamics.
  2. Call for Comprehensive Approaches: Suggest the need for comprehensive approaches in future public health strategies that consider multiple factors, including but not limited to NPIs, vaccinations, and seasonality.

Ethical and Responsible Reporting

  1. Avoid Overstating Conclusions: Be cautious about overstating the conclusions, particularly regarding the effectiveness of NPIs and vaccinations, as these are complex issues with multiple influencing factors.
  2. Transparency in Limitations: Reiterate the limitations of your study to ensure that the conclusions are not misinterpreted or applied beyond their scope.

Overall Structure

  1. Organize for Impact: Structure the conclusions and recommendations to highlight the significance of your findings while providing actionable suggestions for policymakers, researchers, and public health officials.

Incorporating these elements will enhance the clarity, balance, and impact of your conclusions and recommendations, making them more valuable for readers and stakeholders.

Yours sincerely

Comments on the Quality of English Language

While the English is essentially well-written, some aspects can improve the manuscript's readership. For instance, the title is slightly lengthy, and there is a need to maintain consistency in terms (e.g., COVID-19, SARS-CoV-2) throughout the introduction and, while technical language is necessary, ensure it remains accessible to readers who may not be specialists in every aspect of your study. Avoid overly complex jargon where possible.

Round 2

Reviewer 1 Report

Comments and Suggestions for Authors

Thank you for the diligent and thorough response to the previous comments. As stated previously, I appreciate the authors' willingness to investigate what can be seen as dogma in pandemic response, creative use of available data sources and diligence in looking at multiple outcomes and data sources.   

With regard to the expressed concern for the construction / ordering of the paper – the authors have responded that due to the length they opted to combine the results and discussion for the convenience of the readers. It remains my perspective that there are reasons beyond convenience to dispassionately present the results and then more passionately present the interpretation. I do think this could also allow the expression of caveats and limitations at one time in the conclusion verses with each variable as they are now. However, I defer to the Editors for their decision and consistency with the norms of the journal. 

As expressed previously, I appreciate that the nature behavior of the virus(es) themselves and the successive introduction of new variants must be accounted for in understanding the progression of the pandemic. I do still struggle with whether referring to the phenomenon as “seasonality” to apply to the virus behavior in the first 1-3 years is the most helpful, but I do appreciate the addition of the variant waves to Figure 7 which was helpful and I understand that the background Swedish seasonality "stood up" to the statistical tests. 

I cannot help but be reminded of our continual challenge in public health in the inability to measure "what didnt happen," that is what deaths in the omicron waves may have looked like without vaccination for example. With this in mind, my perspective continues to be that the summary statements may be too sweeping, and lack suggestions or implications for public health practice -- for example if seasonality is the most explanatory factor does this suggest that authorities should alter the timing of NPIs or vaccination? Given the heavy weight of the implications for saying that neither NPIs or vaccination was effective at the national level, it seems that there is a responsibility to ensure statements will say only what the author intend when taken out of context. 

Editorial points: Line 103: The change to “COVID 19” on this line to refer to the virus seems problematic as COVID-19 is the illness (or the infection) not the pathogen, despite popular use. I would argue that SARS-CoV-2 is preferred for line 103 while the “COVID-19” terminology in the rest of the paragraph is appropriate.

Edit at lines374- 375 (p 11) is confusing, could this be a typo? (“It should be apparent that all of the commonly-used metrics for the progression of the pandemic have 29 and problems [77]. ”)

Check editing on lines 591-592, line 592 is not a complete sentence: “So that by mid-2021, the opportunity to receive the vaccine was available for all adults.” (while appreciating the explanatory detail.) 

Check edits on line 960 (delete “a”).

The quote from Shamsa et al in lines 990-1 is not clear. Is it possible that this wording is incorrect? 

Reviewer 2 Report

Comments and Suggestions for Authors

The authors have achieved a significant improvement in the quality and scientific rigor of their article. Although positive advances have been made, I still have some minor observations.

Regarding the methodology of the abstract, it is advisable to specify the type of study design used. In addition, I suggest modifying the structure of the article so that the methodology section is presented separately from the introduction. For example, the layout of the article could be as follows: Introduction, Methodology (including details on study design, data, statistical analysis, study area, and COVID-19 pandemic progression for each country), Results, Discussion, Conclusions, and Recommendations.
